# Evaluation of dihydropyranocoumarins as potent inhibitors against triple-negative breast cancer: An integrated of in silico, quantum & molecular modeling approaches

Abu Yousuf Hossin[1*©], Md Naziur Rahman [1©], Md Mahabub Hasan[1], Ansarul Karim [1], Shyikh Ahmed Alif[1], Naziat Nayel Arshi[1], Shammi Jahan[1], Ajoy Kumer [2,3*]

1 College of Agricultural Sciences, IUBAT—International University of Business Agriculture and Technology, Dhaka, Bangladesh, 2 Center for Global Health Research, Saveetha Medical College and Hospitals, Hospitals, Saveetha Institute of Medical and Technical Sciences, Chennai, Tamil Nadu, India, 3 Department of Chemistry, College of Arts and Sciences, IUBAT—International University of Business Agriculture and Technology, Dhaka, Bangladesh

☯ Indicates the equal contribution in this article.
* yousuf.iubat@gmail.com (AYH), kumarajoy.cu@gmail.com (AJ)

## Abstract

Triple-negative breast cancer (TNBC) characterizes one of the most antagonistic subtypes of breast hostilities, due to lacking targeted and potential therapies. In this study, it is investigated and performed an integrative *in silico* investigation into the pharmacological potential of a series of dihydropyranocoumarins (Visnadine (L01), Pteryxin (L02), Isosamidin (L03), and Suksdorfin (L04)) as forthcoming TNBC therapeutics. A many-sided computational workflow was employed encompassing quantum chemical calculations, drug-likeness profiling, *in silico,* molecular docking, and molecular dynamics (MD) simulations. First of all, all ligands were sourced from the PubChem database and performed the geometry optimization with calculating the quantum descriptors using Density Functional Theory (DFT) via the DMol³ module in BIOVIA Materials Studio, applying the B3LYP functional and DNP basis set. Frontier molecular orbital (FMO) analyses in terms of HOMO–LUMO energy gaps and associated global reactivity descriptors, were evaluated to ascertain electronic stability and reactivity trends. Subsequent, PASS prediction, drug-likeness and ADMET assessments, performed using way2drug, SwissADME and pkCSM platforms, revealed favorable pharmacokinetic profiles, with all candidates exhibiting high gastrointestinal absorption, acceptable aqueous solubility, and minimal cytochrome P450 inhibition. Next, Target-based molecular docking against key TNBC-related proteins (PDB IDs: 5HA9 and 7L1X) was conducted using AutoDock Vina within PyRx. These complexes were further validated through 100-ns all-atom MD simulations using Desmond software under the AMBER14 force field, demonstrating stable RMSD values and compact, persistent protein–ligand interactions throughout the simulation period. For

**Data availability statement:** All data supporting the findings of this study are available in the manuscript and the supplementary information file.

**Funding:** The author(s) received no specific funding for this work.

**Competing interests:** The authors have declared that no competing interests exist.

resulting, to begin with, PASS prediction suggested high probabilities for antineoplastic activity, substantiating their biological relevance, and initially, it was revealed that the ligands showed the anti-cancer properties. Next, despite predicted hepatotoxicity, the compounds showed no AMES mutagenicity or genotoxicity, indicating an acceptable safety profile. Overall, dihydropyranocoumarins, especially L03, emerge as promising TNBC leads. Docking analysis revealed strong binding affinities across the ligand set, with Isosamidin (L03) showing the most pronounced interaction (−9.1 kcal/mol), primarily mediated through hydrogen bonding and π-stacking interactions within the active sites. On based on quantum calculation, among the derivatives, L01 exhibited the highest chemical stability, while L04 showed greater electrophilic reactivity, as reflected in the MEP surface and charge distribution profiles. Lastly, the MD indicates indicative of strong conformational stability under physiological conditions of docked complex by RMSD, RMSF, SASA, H bonding and interactions. However, these in silico studies and computational approaches warrant to the future scope for experimental validation through in-vitro assays such as MTT cytotoxicity, apoptosis induction, and cell migration studies to confirm the anti-TNBC potential of these compounds.

## 1. Introduction

Breast cancer is one of the leading causes of cancer-related deaths among women worldwide [1]. The increasing prevalence of this disease presents a significant challenge to the global public health community [2]. In 2020, breast cancer accounted for 11.7% of all new cancer cases worldwide [3]. As the second most commonly diagnosed cancer in women, it remains the foremost cause of cancer mortality in women worldwide [4]. Therefore, TNBC is an aggressive and invasive form of breast cancer that accounts for 15% of all invasive breast cancer cases [5]. Unlike other types, TNBC lacks receptors for estrogen (ER) and progesterone (PR), and it does not have higher than normal levels of the HER2 protein. This absence of receptors and proteins means TNBC grows and spreads faster, has fewer treatment options, and tends to have a worse prognosis. Consequently, it is more challenging to treat and has a higher likelihood of recurrence compared to other breast cancer types [6]. According to epidemiological data, TNBC accounts for roughly 15–20% of all cases of breast cancer and primarily affects premenopausal women under 40 [7]. With a mortality rate of 40% within the first five years after diagnosis, TNBC patients have a shorter survival time than patients with other breast cancer subtypes [8]. About 46% of patients with TNBC experience distant metastases, indicating the disease's extreme aggressiveness. Recurrence rates after surgery can reach 25%, and the median survival time following metastases is only 13.3 months. Furthermore, up to 75% of TNBC patients die within three months of recurrence [9].

Therefore, TNBC has fewer treatment options, is more likely to recur and spread, and has a worse prognosis compared to other types of breast cancer. This is due to the absence of ER, PR, and HER2 expression, rendering hormone and

HER2-targeted therapies ineffective. Consequently, chemotherapy is the main treatment for TNBC; however, this treatment does not significantly extend patients' overall survival time [10,11]. Despite recent advancements and efforts to use targeted therapies for managing TNBC, the high rates of morbidity and mortality due to metastasis to major organs and drug resistance remain significant challenges [12]. Therefore, there is an urgent need to discover effective molecular targets and new therapeutic strategies for treating TNBC.

Natural products offer a versatile border in TNBC treatment, linking traditional medicine and modern oncology. Their integration into suppression-centric strategies could transform TNBC management, offering safer and more effective alternatives to conventional therapies [13,14] Currently, over half of anticancer drugs are derived from natural products or their derivatives, with numerous key chemotherapeutic drugs sourced from plants [15]. Essentially, plant-based bioactive compounds provide an appreciated source of potential chemicals for developing new cancer treatments [12].

The genus *Peucedanum* (family Apiaceae), which comprises over 120 species, is distributed throughout Asia, Africa, and Europe [13]. Conventionally, many of important species within this genus have been used in the treatment of a variety of health conditions [16]. A wide range of phytochemical surveying on *Peucedanum* species have extracted and isolated of various bioactive compounds, including dihydropyranocoumarins and essential oils [17–19]. Among of Dihydropyranocoumarins, it contains a diverse group of plant secondary metabolites, which is characterized by a benzopyrone core structure, that are generally classified into four main subtypes, such as simple dihydropyranocoumarins, pyrone-substituted coumarins, furanocoumarins and pyranocoumarins. Within the *Peucedanum* genus, numerous angular-type pyranocoumarins, such as 8,8-Dimethylpyrano[2,3-f]chromen-2-one, known as fused pyranocoumarin structure, are commonly found, and are supposed to contribute significantly to the pharmacological activities associated with these plants [20,21]. A surveying among the various *Peucedanum* species, *P. praeruptorum*, *P. decursivum*, and *P. japonicum* Thunb (PJT), it has been the most extensively investigated that PJT is a wild herbal plant which commonly found along coastal and cliffside regions near the sea in Native to China and southern Japan. Traditionally, both its aerial parts and roots have been exploited in herbal medicine, while its leaves have also been consumed as a leafy vegetable and garnish in various part of in Okinawa, Japan. In recent years, PJT leaves have gained as vast popularity as a health-promoting food in Japan. In addition, scientific research has increasingly decorated the plant's pharmacological potential: its antioxidant [22], antiplatelet [23], antimicrobial [24], anti-tyrosinase [25], antidiabetic [26] and antirheumatic activities [27]. Secondly, the Pteryxin is another natural coumarin compound which is found in *Peucedanum japonicum* Thunb leaves, recognized for its anti-obesity properties, and it can also play the vital role as a potent butyrylcholinesterase (BChE) inhibitor. MedChemExpress recent reported its pharmacometabolic effects, particularly in neurological contexts, such as its impact on pentylenetetrazole-induced seizures in zebrafish models, where it was observed to influence vagus nerve stimulation [28]. Isosamidin shows strong cytoprotective and antioxidant effects by reducing reactive oxygen species (ROS) generation and modulating apoptotic pathways, a compound linked to diabetic vascular complications. It also inhibits activation of stress-related kinases, for instance p38 and JNK MAPKs, further defending against endothelial dysfunction [29]. Suksdorfin has been reported to promote adipocyte differentiation and improve glucose metabolism abnormalities via activation of PPARγ, suggesting potential benefits in metabolic regulation [30].

Numerous in vivo and in vitro studies have investigated the metabolic pathways of dihydropyranocoumarins [31,32]. However, their potential application in TNBC remains unexplored, particularly through computational drug discovery approaches. Despite growing pharmacological interest in *Peucedanum japonicum* Thunb. and its coumarin derivatives, no study to date has systematically evaluated dihydropyranocoumarins against TNBC using *in silico* methods. Specifically, their interactions with clinically relevant TNBC targets such as PARP1 and CK2α remain uninvestigated. This lack of mechanistic insight represents a critical gap in the literature, which this study aims to address by identifying potential anti-TNBC candidates from PJT-derived natural compounds.

Based on emerging molecular insights into TNBC, we selected two biologically significant protein targets Poly (ADP-ribose) polymerase-1 (PARP1; PDB ID: 5HA9) and Casein kinase 2 alpha (CK2α; PDB ID: 7L1X) for our in-silico

investigations, both of which play pivotal roles in the pathogenesis of triple-negative breast cancer (TNBC). The 5HA9 structure represents PARP1 in complex with a novel antagonist and provides an ideal model for evaluating small-molecule inhibition. PARP1 is a key enzyme in the base excision repair pathway and is known to be overactive in BRCA1/2-mutated TNBC, where the loss of homologous recombination repair creates a synthetic lethality opportunity [33,34]. Thus, PARP1 inhibition has emerged as a clinically validated strategy in the treatment of TNBC, especially in patients harboring BRCA mutations. Meanwhile, 7L1X corresponds to the human CK2α kinase, a serine/threonine protein kinase involved in multiple oncogenic signaling pathways, including PI3K/AKT and NF-κB. Overexpression of CK2α has been documented in aggressive breast cancers, including TNBC, where it contributes to cell survival, proliferation, and metastasis. Targeting CK2α has therefore gained traction as a promising therapeutic strategy [35,36]. By selecting these two targets, one involved in DNA repair and the other in cell survival, we aimed to identify dual-acting or selective dihydropyranocoumarin derivatives that can disrupt critical TNBC survival pathways.

In this study, advanced computational methods were paid to demeanor countless accountabilities that aimed at validating and establishing dihydropyranocoumarins and its derivatives as potential therapeutic agents against TNBC. One of the key and foremost benefits of computational methods is their ability to significantly reduce the time, cost, and resources typically compulsory in drug design and discovery. Moreover, these techniques can enhance the efficiency of the development process by diminishing the likelihood of failure during pre-clinical or clinical trials [37]. In contrast, despite limited scientific data and lack of research resources on the biological activities of dihydropyranocoumarins in various metabolic diseases, it makes the scope of investigating their effects on TNBC. Therefore, this study expected to investigate the potential anti-TNBC activity of dihydropyranocoumarin derivatives using advanced *in silico* techniques. To the best of our knowledge, this is the first comprehensive *in silico* investigation of dihydropyranocoumarin derivatives from *Peucedanum japonicum* Thunb targeting TNBC-associated proteins PARP1 (5HA9) and CK2α (7L1X).

## 2. Materials and methodology

### 2.1. Preparation of ligand

For the preparation of ligands, the entire chemical structures from PubChem (https://pubchem.ncbi.nlm.nih.gov/). The dihydropyranocoumarins derivatives were optimized using DFT via the DMol³ module in BIOVIA Materials Studio, employing DNP basis set. Default convergence criteria ensured accurate geometries for further quantum and molecular analyses [14,38,39]. The B3LYP functional jointly with DNP basis sets was utilized within the DMol³ code framework to achieve accurate computational results. These analytical methods were employed to regulate the frontier molecular orbitals namely, the highest occupied molecular orbital (HOMO) and the lowest unoccupied molecular orbital (LUMO) along with their respective amplitudes following geometry optimization. The optimized molecular structures were exported in PDB format for *in silico* and computational analyses, which are named as molecular docking, molecular dynamics (MD) simulations, and ADMET profiling. Key chemical reactivity parameters, for example energy gap, hardness, electronegativity, affinity and softness, were estimated using established and validated algorithms.

### 2.2. Determination of PASS prediction

The pass prediction data (Pa > Pi value) was obtained from the steadfast and valued website http://way2drug.com/PassOnline/predict.php. This value is essential for guessing the antiviral, antifungal, anticancer, antibacterial, and antibiotic properties of molecules. It plays a crucial role in evaluating and selecting the therapeutic and biological potency of new drug compounds [40].

### 2.3. Lipinski rule and pharmacokinetics

Lipinski's Rule of Five is a guideline used to evaluate the drug-likeness of a chemical compound, particularly its potential to be an orally active drug in humans. This rule considers many key factors, for instance the number

of hydrogen bond acceptors, hydrogen bond donors, topological polar surface area (TPSA), and bioavailability score and many more parameters. These properties support to determine whether a compound has the necessary chemical and physical attributes for oral bioavailability. The SwissADME website (http://www.swissadme.ch/index.php) was used and for determination the valuable resource for assessing these parameters, offering a free platform to analyze, and predicted the drug-likeness of chemical compounds based on Lipinski's criteria [41,42].

## 2.4. Determination of the data of ADMET

Developing new drugs often faces high catastrophe charges due to insufficient pharmacokinetic and security profiles. Computational techniques and in silico can help to mitigate these issues. One promising tool for foreseeing pharmacokinetic characteristics is pkCSM, which focuses on ADMET (Absorption, Distribution, Metabolism, Excretion, and Toxicity) features. ADMET features were assessed using the ADMETSAR online database, which is considered highly reliable and ideal for predicting ADMET parameters [43,44]. The ADMETsar database could be accessed - http://lmmd.ecust.edu.cn/admetsar1/ in the date of 21 March 2025.

## 2.5. Preparation of protein

For this study, triple-negative breast cancer answerable protein (PDB ID: 5HA9) and (PDB ID: 7L1X) [45,46] were found and collected from the RCSB Protein Data Bank (https://www.rcsb.org/). The selected protein was chosen due to their established involvement in TNBC-related DNA repair and oncogenic signaling pathways, respectively, as elaborated in the introduction. These structures often lack whole connectivity information, bond orders, and formal charges, making thorough pre-processing indispensable for precise structure preparation. A comparison of their structural quality parameters, and essential properties has summarized in Table 1 and Fig 1.

First, these raw protein structures were imported into the Discovery Studio software, where water molecules, unusual ligands, and heteroatoms were removed to prepare and optimize the proteins by tools of software. As a result, clean protein structures were obtained that were then exported as PDB files. Following this cleaning process, energy minimization was completed to resolve any steric clashes or unrealistic geometries, ensuring the protein reached a stable conformation without meaningfully changing its overall structure.

The stereochemical quality of the selected protein structures (PDB IDs: 5HA9 and 7L1X) was evaluated using PROCHECK, which generates Ramachandran plots added in S1 Fig. For 5HA9, 86.9% of residues were located in the most favored regions and 12.9% in the additionally allowed regions, with no residues in generously allowed or disallowed regions S1 Fig. For 7L1X, 91.3% of residues were located in the most favored regions and 8.0% in the additionally allowed regions, with no residues in generously allowed or disallowed regions S1 Fig. These results confirm the high stereochemical quality of both protein structures, supporting their suitability for docking studies.

**Table 1. Comparison of structural quality parameters for two protein targets (PDB IDs: 7L1X and 5HA9) from *Homo sapiens*.**

| Title | PDB ID: 7L1X | PDB ID: 5HA9 PDB ID: 5HA9) |
|---|---|---|
| Organism | *Homo sapiens* | *Homo sapiens* |
| Resolution | 1.80 Å | 4.01 Å |
| R-Value Free | 0.221 | 0.358 |
| Ramachoron plot, % | 95 | 98 |
| References | [47] | [48] |

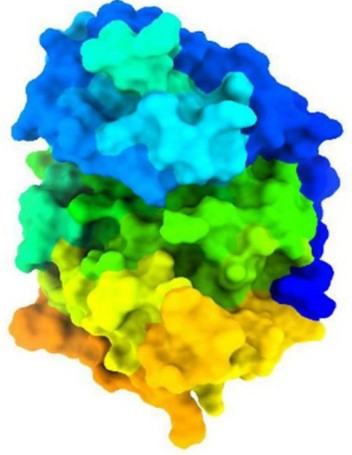 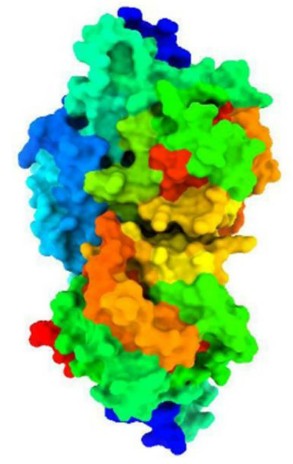

**Human CK2 alpha kinase (PUB ID 7L1X)**          **TNBC receptor (PUB ID 5HA9)**

**Fig 1. Three-dimensional protein structure of TNBC.**

## 2.6. Procedure of molecular docking

The cleaned and optimized proteins were exported as PDB files for further computational investigations. Molecular docking was performed using the PyRx program with AutoDock Vina [49], utilizing grid box parameters (detailed in Table 2). The docked compounds were then analyzed and visualized using Discovery Studio [50]. Subsequent analyses included molecular dynamics and ADMET (absorption, distribution, metabolism, excretion, and toxicity) studies to comprehensively evaluate protein-ligand interactions.

## 2.7. Optimization and ligand preparation

Computational modeling and molecular optimization were carried out using Material Studio 8.0, a widely used software for materials simulation and quantum chemical calculations. The DMol3 module, an advanced DFT quantum mechanical code, was employed for precise determination of chemical descriptors, quantum properties, and geometrical optimization of the molecules studied. To ensure accuracy and reliability in the calculations, the double numerical polarization (DNP) basis set was utilized in conjunction with the DFT functional for geometry optimization and evaluation of quantum descriptors calculations [51–53]. The computed quantum properties included HOMO (highest occupied molecular orbital), LUMO (lowest unoccupied molecular orbital), energy gap, ionization potential, electron affinity, chemical potential, electronegativity, hardness, softness, and the electrophilicity index using equation from 01 to 08. These descriptors provided valuable

**Table 2. Grid box parameters used for PyRx docking analysis in this study.**

| Protein Name with PDB ID | Grid Box Size (PYRX) | |
|---|---|---|
| | Center | Dimension(Å) |
| *Homo sapiens* **PDB ID: 7L1X** | X = 10.2690 | X = 65.5753 |
| | Y = −0.0137 | Y = 53.4614 |
| | Z = 15.0250 | Z = 54.1291 |
| *Homo sapiens* **PDB ID: 5HA9** | X = −3.9220 | X = 83.0519 |
| | Y = 11.3294 | Y = 72.2407 |
| | Z = −21.7292 | Z = 89.2767 |

insights into molecular stability, electronic structure, and reactivity, supporting the assessment of their potential applications in various chemical and material science domains.

$$E_{gap} = (E_{LUMO} - E_{HOMO}) \tag{1}$$

$$I = -E_{HOMO} \tag{2}$$

$$A = -E_{LUMO} \tag{3}$$

$$(X) = \frac{I + A}{2} \tag{4}$$

$$(\omega) = \frac{\mu^2}{2\eta} \tag{5}$$

$$(\mu) = -\frac{I + A}{2} \tag{6}$$

$$(\eta) = \frac{I - A}{2} \tag{7}$$

$$(S) = \frac{1}{\eta} \tag{8}$$

### 2.8. Molecular Dynamics Simulations (MDs)

The primary objective of molecular dynamics (MD) simulations was to assess the stability of chemical compounds within drug–protein complexes which was docked after docking procedure, and it was the one the validation procedure of docking. This simulation was performed using the Desmond MD engine (Schrödinger, LLC), which suggests high-performance scalability and efficient GPU acceleration for large biomolecular systems. The simulation trajectories were done for 100 nanoseconds (ns) under an NPT ensemble.

The protein–ligand complex (holoform) was prepared using the Protein Preparation Wizard in Maestro (Schrödinger Suite), where missing side chains and loops were modeled and rebuilt, bond orders assigned, and hydrogen atoms added. The protonation states of ionizable residues and ligands were determined at physiological pH (7.4) using Epik and PROPKA, ensuring accurate assignment of charges or polar charge of atoms. The OPLS4 force field was pragmatic to use to both the protein and the ligand ensuring for the consistent parameterization and reliable interaction modeling.

The system was solvated in an explicit TIP3P water model using a cubic simulation box with a buffer distance of 20 Å around the complex. Next, 0.15 M NaCl was added to mimic physiological ionic strength, and periodic boundary conditions were applied in all directions. Prior to the production run, the system underwent energy minimization and a multi-step equilibration protocol as recommended by Desmond's default relaxation procedure.

Upon completion of the simulations, structural and dynamic parameters such as root mean square deviation (RMSD) and root mean square fluctuation (RMSF), Radius of Gyration (Rg), Total Energy and Potential Energy, were calculated using the Simulation Interaction Diagram (SID) and Maestro Analysis tools. In addition, Protein–Ligand Interactions, such as hydrogen bonds, hydrophobic contacts, π–π stacking, salt bridges, water bridges were evaluated for giving more strong evidence to the conformational stability and flexibility of the protein–ligand complex.

## 3. Results and discussions

### 3.1. Chemistry and DFT study

**3.1.1. Optimized structure and their chemistry.** The molecular and geometry optimization of L01, L02, L03, and L04 using DFT calculations offer understandings into their structural stability, electronic properties, and chemical reactivity. The optimized geometrized structure conveyed expected C–C and C–O bond lengths, with C=O bonds being quicker due to electron delocalization. The lactone and furan rings exhibited near-planar geometries, contributing to conjugation, whereas the L01 and L03 display higher angular strain that is caused for additional substitutions. The HOMO-LUMO energy gap study indicates that L04 has the smallest energy gap, suggesting higher reactivity, while L01 has the largest value, demonstrating greater chemical and physical stability. The HOMO orbitals are primarily localized over oxygenated furanocoumarin rings, while LUMO orbitals are concentrated in electrophilic regions, influencing charge transfer interactions. Dipole moment analysis showed that L01 and L03 have higher polarity, favoring polar interactions, while L02 and L04 are more soluble in nonpolar environments. Charge distribution analysis revealed oxygen atoms in the lactone rings carried significant negative charge, enhancing potential interactions with electrophilic species. These results highlight how their electronic and geometric characteristics impact their use in pharmaceutical, catalytic, and material science applications, indicating the need for further study into their functional roles using approaches like molecular docking and spectroscopic validation. These derivatives' optimized chemical structures are shown in Fig 2.

Chemical reactivity and kinetic stability of a molecule are primarily determined by its frontier molecular orbitals, namely the highest occupied molecular orbital (HOMO) and the lowest unoccupied molecular orbital (LUMO). The transition of an electron from HOMO to LUMO governs molecular excitation, requiring a significant energy input. The HOMO-LUMO energy gap is a key indicator of kinetic stability and reactivity, with a larger gap correlating with increased stability and reduced reactivity.

Table 3 presents the computed molecular orbital energies along with key chemical descriptors, including energy gaps, chemical potential, electronegativity, hardness, and softness. Among the ligands studied, L01 exhibited the largest HOMO-LUMO energy gap (7.201 eV), indicating its higher stability. In contrast, ligand L02 had the highest softness value (0.3126), suggesting greater reactivity and solubility. The highest hardness value (3.604) was observed for ligand L04, signifying stronger resistance to deformation and greater stability under physiological conditions [54–56].

Hard ligands (higher η) are less polarizable and form strong, stable interactions such as hydrogen bonds and electrostatic contacts, but their rigidity limits adaptability. In this study, L01 and L04 (η ≈ 3.60) are relatively harder, suggesting stronger binding but reduced flexibility. Soft ligands (higher σ) are more polarizable, adaptable, and often show better permeability. L02, with the highest softness (σ ≈ 0.313), may engage in versatile non-covalent interactions and diffuse more readily through membranes, though with weaker binding [57,58].

Overall, effective drug candidates require a balance: hardness for stable target binding and softness for permeability. Here, L01/L04 may provide strong interactions, while L02 offers better adaptability and bioavailability.

Chemical potential plays a crucial role in defining molecular stability and reactivity, influencing interactions under standard thermodynamic conditions. Variations in these parameters highlight significant differences in the stability and reactivity of the studied ligands, providing valuable insights into their potential applications.

**3.1.2. Frontier Molecular Orbitals (HOMO and LUMO).** Frontier molecular orbitals (FMOs), specifically the HOMO and the LUMO, play a crucial role in determining the chemical reactivity and stability of small molecules. These orbitals

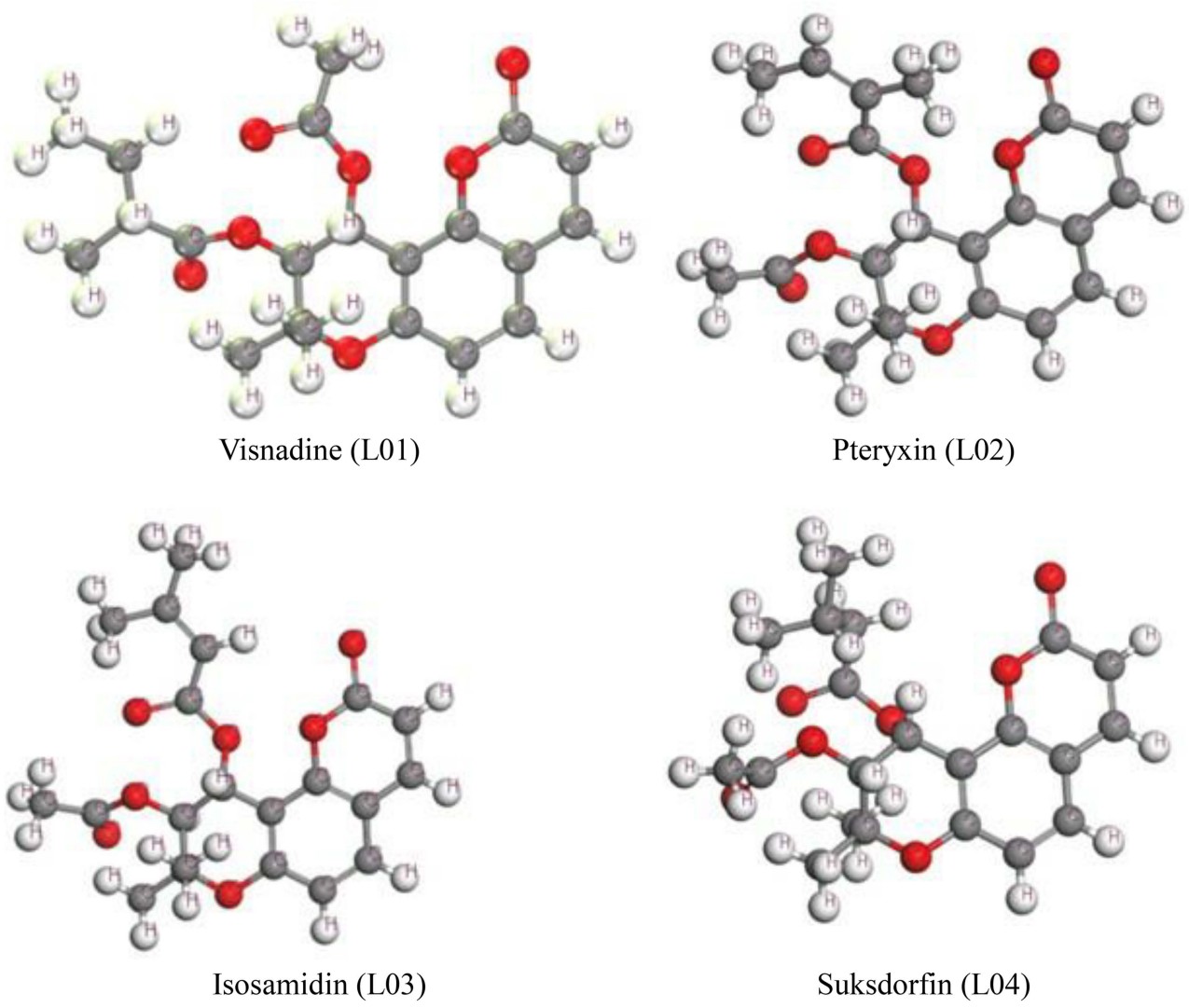

Visnadine (L01)          Pteryxin (L02)

Isosamidin (L03)         Suksdorfin (L04)

**Fig 2. Optimized Structure of four ligands** *Frontier Molecular Orbitals and Chemical Reactivity Descriptors.*

**Table 3. Data of chemical descriptors calculation.**

| Ligand | LUMO | HOMO | A=-LUMO | I=-HOMO | Energy gap=I-A | Chemical Potential ($\mu$)= -I+A/2 | Hardness ($\eta$)= I-A/2 | Electronegativity (x)=I+A/2 | Softness ($\sigma$)=1/n | Electrophilicity ($\omega$)=$\mu$2/2$\eta$ |
|---|---|---|---|---|---|---|---|---|---|---|
| L01 | −2.003 | −9.204 | 2.003 | 9.204 | 7.201 | −5.6035 | 3.6005 | 5.6035 | 0.2777 | 4.3604 |
| L02 | −1.394 | −7.791 | 1.394 | 7.791 | 6.397 | −4.5925 | 3.1985 | 4.5925 | 0.3126 | 3.2970 |
| L03 | −1.85 | −9.048 | 1.850 | 9.048 | 7.198 | −5.4490 | 3.5990 | 5.4490 | 0.2779 | 4.1250 |
| L04 | −2.056 | −9.264 | 2.056 | 9.264 | 7.208 | −5.6600 | 3.6040 | 5.6600 | 0.2775 | 4.4445 |

help identify chemically active bonds and influence catalytic properties. The transition of an electron from the HOMO to the LUMO occurs during electronic absorption, and their energy levels can be accurately calculated using DFT. The energy gap between HOMO and LUMO serves as an indicator of chemical stability larger gaps suggest greater stability, whereas smaller gaps indicate lower stability and increased reactivity.

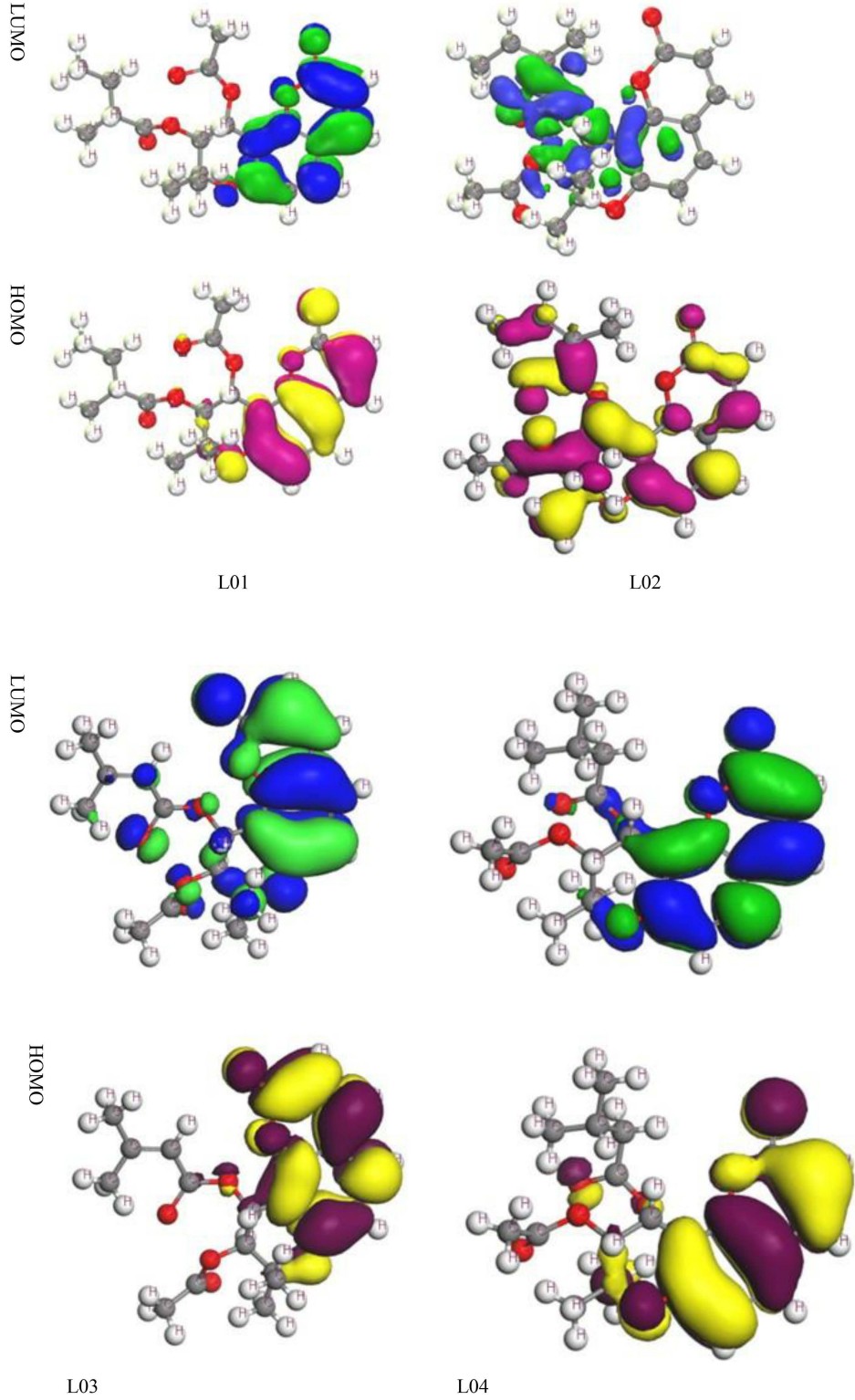

**Fig 3. Molecular Frontier Orbital for HOMO and LUMO.**

The frontier molecular orbital (FMO) analysis of the dihydropyranocoumarin derivatives revealed distinct distributions of electron density, with the HOMO localized primarily over the aromatic coumarin core, while the LUMO extended across the pyran ring and substituent regions Fig 3. This spatial separation suggests efficient intramolecular charge transfer upon excitation, a feature that enhances molecular reactivity. The reddish-brown regions in the visualized orbitals correspond to electron-rich areas (positive phase), whereas the green regions denote electron-deficient zones (negative phase).

Compared with previously reported coumarin-based derivatives, the observed HOMO–LUMO gaps of our dihydropyranocoumarins are moderately narrower, indicating improved electron delocalization and enhanced reactivity. For instance, similar studies on furanocoumarins and benzopyran derivatives have shown larger HOMO–LUMO gaps, reflecting lower charge-transfer efficiency and reduced bioactivity potential. In contrast, our derivatives exhibit higher frontier orbital overlap, suggesting stronger interaction potential with biomolecular targets, which aligns with their predicted anticancer activity [59–61].

The electronic distribution patterns also support the binding behavior observed in molecular docking: compounds with higher HOMO density near functional substituents displayed stronger hydrogen bonding with protein residues, while those with delocalized LUMOs facilitated π–π stacking and hydrophobic contacts. Taken together, the FMO analysis not only validates the molecular stability of dihydropyranocoumarins but also highlights their superior electronic reactivity compared with other related natural scaffolds, underscoring their promise as drug-like candidates [57,58].

HOMO regions typically indicate areas of high electron density, where electrophiles are likely to interact, often extending to functional groups such as hydroxyls due to the presence of electronegative oxygen atoms. In contrast, LUMO regions suggest electron-deficient areas, where nucleophilic interactions are more favorable. Understanding these molecular orbital configurations is essential for predicting chemical reactivity, biological interactions, and electronic properties.

**3.1.3. Electrostatic potential map for charge distribution.** The electrostatic potential map (MEP) is a crucial tool for analyzing molecular interactions and assessing the reactivity of chemical species. It provides insights into charge distribution, which is essential in studying molecular binding mechanisms, particularly in ligand-protein interactions. MEP is widely used in theoretical investigations to explore the physical, biological, and chemical properties of molecules by identifying active sites of molecular orbitals.

From Fig 4, the electrostatic potential charge distribution of the studied molecules is visualized using color-coded isosurfaces. The red regions indicate areas of negative charge, while the blue regions correspond to positive charge distributions. The molecular structures labeled L01, L02, L03, and L04 exhibit varying charge concentrations, suggesting differences in electronic properties. The charge distribution ranges from approximately −9.679 to 4.666, as indicated by the color scale in the Fig 3. The interior regions of the molecules, particularly within the ring structures, predominantly exhibit positive potential (blue regions), whereas the outer regions, especially around oxygen-containing functional groups, show negative potential (red regions).

## 3.2. In-silico study

**3.2.1. PASS prediction spectrum.** The PASS online platform (http://way2drug.com/PassOnline/predict.php) [62] was utilized to assess the predicted biological activities of the studied compounds. This tool aids in identifying potential therapeutic properties that may later be confirmed through experimental studies. PASS prediction evaluates a compound's likelihood of exhibiting specific biological activities by comparing its structure to that of known biologically active substances. This approach is valuable at the early stages of drug discovery and development. The optimized molecular structures of the target compounds were submitted in MOL file format to the PASS online system, which predicted their potential mechanisms of action and bioactivities. These predictions are based on the probability of activity (Pa) and inactivity (Pi). By applying PASS algorithms and filters, researchers can efficiently screen large libraries of drug candidates, focusing on those with the highest potential, thereby saving time and resources. This strategy can significantly accelerate the drug discovery process and improve the chances of identifying effective treatments for various diseases

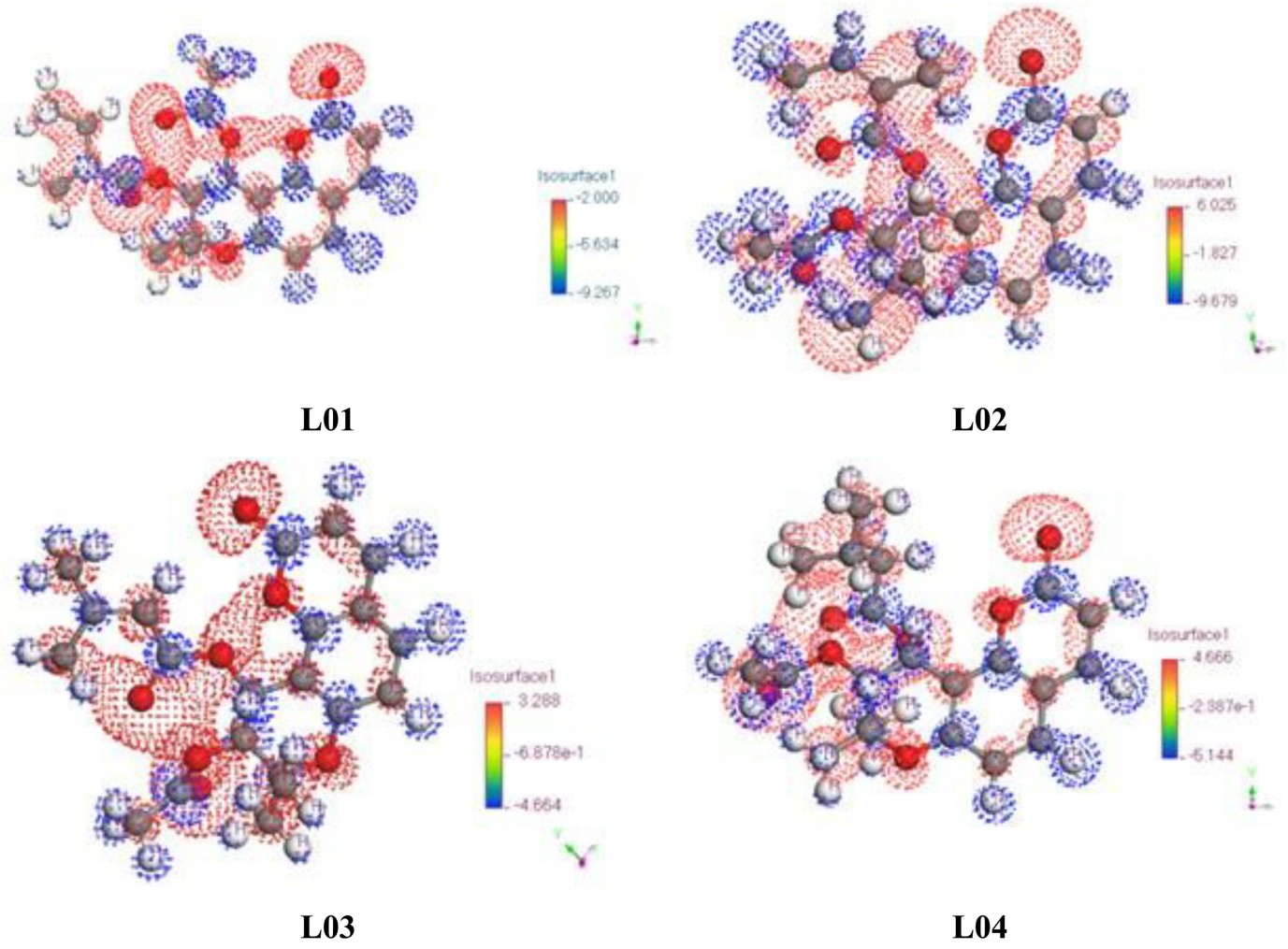

**L01**

**L02**

**L03**

**L04**

**Fig 4. Electrostatic potential map for Frontier Molecular Orbital.**

[62,63]. The PASS prediction spectrum reveals significant insights into the potential bioactivities of the compounds studied. According to established PASS prediction guidelines, a Pa value > 0.7 is generally considered highly confident, indicating a strong likelihood of biological activity. Values between 0.5 and 0.7 are considered moderate, while those below 0.5 are typically viewed as low-confidence predictions and less likely to be experimentally confirmed. Regarding data, the probability of being active (Pa) value range is 0.372 to 0.392 for Antiviral, 0.480 to 0.539 for Antibacterial, 0.485 to 0.589 for antifungal, the last one is antineoplastic, which is 0.709 to 0.893 showing in (Table 4). Among all predicted activities, antineoplastic activity yielded the highest Pa values, highlighting these compounds' stronger potential as anticancer agents relative to their antimicrobial or antiviral roles. Notably, the Pa values associated with antineoplastic activity are substantially higher compared to those for antiviral, antibacterial, and antifungal properties. All four compounds demonstrated high confidence levels for antineoplastic activity, with Pa values ranging from 0.709 to 0.893, thereby reinforcing their selection for further anticancer evaluation. Based on this finding, subsequent analyses were directed toward investigating their therapeutic potential against triple-negative breast cancer. Consequently, two proteins associated with triple-negative breast cancer were selected as target receptors for further computational analysis.

**Table 4. Computational data of PASS Prediction Activity spectrum.**

| Ligand | Antiviral | | Antibacterial | | Antifungal | | Antineoplastic | |
|---|---|---|---|---|---|---|---|---|
| | Pa | Pi | Pa | Pi | Pa | Pi | Pa | Pi |
| L01 | 0.387 | 0.016 | 0.480 | 0.018 | 0.553 | 0.023 | 0.709 | 0.025 |
| L02 | 0.372 | 0.018 | 0.539 | 0.013 | 0.589 | 0.020 | 0.893 | 0.005 |
| L03 | 0.392 | 0.015 | 0.512 | 0.015 | 0.485 | 0.033 | 0.857 | 0.006 |
| L04 | 0.383 | 0.017 | 0.494 | 0.017 | 0.485 | 0.033 | 0.717 | 0.023 |

However, it is important to recognize the limitations of PASS predictions. These models are based on structure–activity similarities and do not capture target-specific interactions, mechanisms of action, or pharmacodynamic complexities. Therefore, false positives may occur, and predictions must be viewed as preliminary hypotheses. Further experimental validation, including biochemical, cellular, or in vivo assays, is crucial to confirm the biological relevance and therapeutic applicability of these predictions.

**3.2.2. Lipinski's rule of five and drug-likeness.** According to Lipinski's Rule of Five, orally active drugs should generally possess certain physicochemical properties, including relatively low molecular weight, to ensure adequate bioavailability in mammals [64]. This rule serves as a guideline for evaluating whether a compound is likely to be orally administered based on its molecular characteristics and biological activity. In the present study, the molecular weights of the predicted compounds ranged from 386.40 to 388.416 Da, and their topological polar surface areas (TPSA) varied between 92.04 Å². Most of the compounds exhibited favorable bioavailability scores 0.55. Gastrointestinal (GI) absorption, another key parameter for oral bioavailability, was generally low among the compounds; however, L01 to L04 all compounds demonstrated high GI absorption rates (Table 5). Overall, all compounds complied with Lipinski's criteria.

To contextualize these findings, a comparative analysis was performed against established TNBC chemotherapeutics such as Paclitaxel, Docetaxel, Doxorubicin, and Epirubicin. These known drugs generally violate multiple Lipinski parameters, particularly due to higher molecular weights (543–853 Da), elevated TPSA, and poor GI absorption (Table 5). In contrast, the PJT ligands showed superior drug-likeness profiles, suggesting more favorable pharmacokinetic behavior and oral bioavailability potential [65]. This comparison highlights the promising nature of these natural product derivatives as candidate leads for TNBC therapy.

**3.2.3. Bioavailability radar analysis.** The bioavailability radar plots S2 Fig. provide a visual summary of six key physicochemical parameters influencing oral drug-likeness: lipophilicity (LIPO), size (SIZE), polarity (POLAR), solubility (INSOLU), saturation (INSATU), and flexibility (FLEX). Compounds whose properties fall within the pink optimal zone are

**Table 5. Comparative analysis of pharmacokinetic and Lipinski parameters between PJT-derived ligands and known TNBC drugs.**

| Ligand | Number of rotatable bonds | Hydro-gen bond acceptor | Hydro-gen bond donor | Topological polar surface area, Å2 | Consen-sus Log Po/w | Lipinski rule | | Molecular weight (g/mol) | Bioavail-ability Score | Gastro-intestinal absorption |
|---|---|---|---|---|---|---|---|---|---|---|
| | | | | | | Result | viola-tion | | | |
| L01 | 6 | 7 | 0 | 92.04 | 3.23 | Yes | 0 | 388.41 | 0.55 | High |
| L02 | 5 | 7 | 0 | 92.04 | 3.08 | Yes | 0 | 386.40 | 0.55 | High |
| L03 | 5 | 7 | 0 | 92.04 | 3.18 | Yes | 0 | 386.40 | 0.55 | High |
| L04 | 6 | 7 | 0 | 92.04 | 3.15 | Yes | 0 | 388.41 | 0.55 | High |
| Paclitaxel | 15 | 14 | 4 | 221.29 | 3.52 | No | 2 | 853.91 | 0.17 | Low |
| Docetaxel | 14 | 14 | 5 | 224.45 | 2.88 | No | 2 | 807.88 | 0.17 | Low |
| Doxoru-bicin | 5 | 12 | 6 | 206.07 | 0.52 | No | 3 | 543.52 | 0.17 | Low |
| Epirubicin | 5 | 12 | 6 | 206.07 | 0.52 | No | 3 | 543.52 | 0.17 | Low |

considered more likely to have good oral bioavailability. All four PJT-derived ligands (L01–L04) were well-positioned within the optimal zone for all six parameters, indicating balanced lipophilicity, appropriate molecular size, moderate polarity, favorable solubility, suitable saturation levels, and optimal flexibility.

In contrast, the known TNBC drugs displayed deviations from the optimal range in multiple parameters. Paclitaxel and Docetaxel exceeded the upper limit for size and FLEX, reflecting their large molecular structures and high conformational flexibility, which may reduce passive oral absorption. Doxorubicin and Epirubicin showed borderline polarity values exceeding the optimal range, potentially limiting their membrane permeability. This comparative analysis reinforces the superior oral drug-likeness profile of PJT-derived ligands, supporting their potential as more bioavailable and pharmacokinetically favorable candidates for TNBC therapy.

**3.2.4. ADME properties studies.** ADME properties are crucial for determining a bioactive molecule's potential, encompassing absorption, distribution, metabolism, and excretion. There are numerous descriptors for ADME, including parameters: human intestinal absorption (HIA), blood-brain barrier (BBB) permeability, Caco-2 permeability, human intestinal absorption, P-glycoprotein inhibitor, P-glycoprotein substrate, renal organic cation transporter, sub-cellular localization, CYP4502C9 substrate, and CYP4501A2 inhibitor, as detailed in Table 6.

The ADME predictions reveal favorable pharmacokinetic properties for all compounds. The water solubility (log S) values, ranging from −4.33 to −4.659, indicate moderate solubility, which is often beneficial for oral drugs. Typically, slightly soluble compounds have log S values between (−4 to −6), while highly soluble compounds have values between (−2 to −4). Based on these solubility parameters, the compounds in question fall within the range of moderate solubility. The Caco-2 permeability values (1.25 to $1.374 \times 10^{-6}$ cm/s) and high intestinal absorption rates (91.282% to 100%) further support their potential for efficient absorption. In terms of distribution, all compounds show a low volume of distribution (log VDss), suggesting limited distribution into body tissues. The BBB (blood-brain barrier) predictions indicate that these compounds are unlikely to cross the BBB, which is advantageous for targeting peripheral cancers while minimizing central nervous system side effects. Regarding metabolism, none of the compounds are predicted to inhibit CYP450 1A2 or 2C9, reducing the risk of drug-drug interactions. The total clearance values (0.754 to 0.95 ml/min/kg) and the absence of renal OCT2 substrate interactions suggest efficient excretion, minimizing the risk of accumulation and toxicity.

When compared with established TNBC therapeutics such as Paclitaxel, Docetaxel, Doxorubicin, and Epirubicin, the PJT-derived ligands (L01–L04) demonstrate superior pharmacokinetic profiles in several key areas. Unlike these reference drugs, which exhibit lower Caco-2 permeability and poor gastrointestinal absorption (e.g., Doxorubicin and Epirubicin: ~62%), the PJT compounds achieve nearly 100% predicted absorption. Moreover, standard agents tend to have

**Table 6. Comparative analysis of ADMET parameters between PJT-derived ligands and known TNBC drugs.**

| Ligand | Absorption | | | Distribution | | Metabolism | | Excretion | |
|---|---|---|---|---|---|---|---|---|---|
| | Water Solubility Log S | Caco-2 Permeability (10−6 cm/s) | Intestinal absorption (human)% | VDss (human) (log L/kg) | B.B.B Permeability | CYP450 1A2 Inhibitor | CYP450 2C9 Inhibitor | Total clearance (ml/min/kg) | Renal OCT2 substrate |
| L01 | −4.33 | 1.25 | 91.282 | −0.436 | No | No | Yes | 0.875 | No |
| L02 | −4.392 | 1.344 | 100 | −0.51 | No | No | Yes | 0.95 | No |
| L03 | −4.392 | 1.344 | 100 | −0.51 | No | No | Yes | 0.861 | No |
| L04 | −4.659 | 1.374 | 100 | −0.47 | No | No | Yes | 0.754 | No |
| Paclitaxel | −3.158 | 0.623 | 100 | 1.458 | No | No | No | −0.36 | No |
| Docetaxel | −3.258 | 0.521 | 100 | 1.163 | No | No | No | −0.411 | No |
| Doxorubicin | −2.915 | 0.457 | 62.372 | 1.647 | No | No | No | 0.987 | No |
| Epirubicin | −2.915 | 0.457 | 62.372 | 1.647 | No | No | No | 0.987 | No |

higher VDss and are associated with poor water solubility (log S ~ −2.9 to –3.2), which may contribute to systemic toxicity. In contrast, PJT ligands show moderate solubility and lack CYP inhibition liability, indicating a lower risk of metabolic side effects. Additionally, all known TNBC drugs in the comparison exhibit low renal clearance and some lack complete ADME data, while PJT compounds offer a completer and more balanced ADMET profile. These findings further support the potential of PJT-derived ligands as safer and more effective drug candidates.

**3.2.5. Aquatic and non-aquatic toxicity.** The toxicity profiles of the compounds were analyzed using several parameters. None of the compounds exhibited AMES toxicity, suggesting they are not mutagenic. The maximum tolerated dose in humans ranged from 0.091 to 0.603 mg/kg/day, with compound 1 being the highest. Oral rat acute toxicity ($LD_{50}$) values were around 3.251 to 3.416 mol/kg, and chronic toxicity values ranged from 1.372 to 2.112 mg/kg/day. Notably, all compounds showed hepatotoxicity, which could limit their clinical use due to potential liver damage. On the other hand, current scientific evidence indicates that PJT, which is widely consumed as a vegetable and traditional medicine in East Asia, does not exhibit liver toxicity. In fact, several studies highlight its safety and potential health benefits. The plant has a long history of being eaten as a vegetable and used medicinally for conditions such as colds, rheumatoid arthritis, and inflammatory diseases, with no reports of liver toxicity in these contexts [66,67]. Research on *P. japonicum* focuses on its anti-inflammatory, antioxidant, and metabolic benefits, including improvements in lipid metabolism and protection against oxidative stress, rather than any adverse hepatic effects [67–69]. In addition, C57BL/6 mice treated with dihydropyranocoumarin-containing PJT did not exhibit any signs of hepatotoxicity, as shown in Table 7 [31]. Aquatic and non-aquatic poisoning are all of the medications listed in Table 7.

All compounds (L01–L04) exhibit poor water solubility, with Log S values ranging from −4.33 to −4.659, indicating limited solubility in aqueous media. Among them, L01 has the highest solubility (−4.33), while L04 is the least soluble (−4.659), which may affect bioavailability, particularly in oral formulations. Caco-2 permeability values range from 1.25 to $1.374 \times 10^{-6}$ cm/s, which falls within a moderate range. Compounds L02–L04 demonstrate slightly better permeability than L01, suggesting improved potential for passive diffusion through intestinal epithelium. All compounds show excellent predicted human intestinal absorption, with values near or at 100%, making them highly favorable for oral delivery.

VDss values range from −0.436 (L01) to −0.51 (L02/L03). These negative values imply that the compounds are likely to stay in the plasma compartment, with low tissue distribution. Lower VDss can be advantageous for targeting blood-borne diseases but may limit efficacy for intracellular or tissue-specific targets. All compounds are predicted to be non-permeable to the BBB, suggesting low risk of central nervous system (CNS) side effects, which is advantageous unless CNS activity is desired.

All compounds are predicted to be non-inhibitors of CYP1A2 and CYP2C9, which is favorable for reducing metabolic drug–drug interactions. Their lack of CYP450 inhibition suggests low metabolic liability and a potentially stable pharmacokinetic profile. Clearance values vary slightly from 0.754 (L04) to 0.95 (L02). These moderate clearance values suggest adequate systemic retention, allowing the compounds sufficient time to exert their biological effect. L02 appears to clear slightly faster than others, which may require consideration in dosing frequency. None of the compounds are predicted to be substrates for the renal organic cation transporter 2 (OCT2), reducing concerns about renal transporter-mediated drug interactions or nephrotoxicity.

**Table 7. Aquatic and non-aquatic toxicity value prediction.**

| Ligand | AMES toxicity | Max. tolerated does (human) mg/kg/day | Oral rate acute toxicity (LD50) (mol/kg) | Oral rat chronic toxicity (mg/kg/day) | Hepatoxicity |
|--------|---------------|----------------------------------------|-------------------------------------------|----------------------------------------|--------------|
| L01 | No | 0.603 | 3.251 | 1.372 | Yes |
| L02 | No | 0.091 | 3.416 | 1.454 | Yes |
| L03 | No | 0.091 | 3.416 | 1.454 | Yes |
| L04 | No | 0.209 | 3.305 | 2.112 | Yes |

### 3.3. Molecular docking

**3.3.1. Binding affinities and molecular interactions.** Molecular docking was employed to simulate the interactions between small molecules and target proteins at the atomic level, providing insights into binding efficiency and enabling the estimation of binding energies within the active sites of target proteins [70–72]. Docking simulations were performed using AutoDock via the PyRx interface, and the binding affinities of the ligands were calculated. A binding energy of −6.0 kcal/mol or lower is typically considered indicative of significant biological activity [73]. Among the dihydropyranocoumarins tested, compounds L02 and L01 demonstrated the highest binding affinities against TNBC receptor (PDB ID: 5HA9), while L03 showed the strongest overall binding potential across all evaluated compounds Table 8.

Among the tested compounds, L03 exhibited the strongest binding affinity (−9.1 kcal/mol) toward CK2 alpha kinase (PDB ID: 7L1X), indicating a highly stable ligand–protein interaction. Notably, this binding occurred without the formation of hydrogen bonds, implying a key role for hydrophobic interactions (six in total), which may involve strong van der Waals forces or π–π stacking within the hydrophobic pocket of the kinase. The binding affinity order for CK2α was: L03 > L04 > L02 > L01. Although both L01 and L04 formed seven hydrophobic bonds, L04 achieved better binding energy (−8.6 kcal/mol), highlighting that binding strength is influenced not only by the number but also the quality and positioning of interactions.

To address this, we redocked a known TNBC inhibitor into the corresponding protein structures (CK2α, PDB ID: 7L1X; TNBC receptor, PDB ID: 5HA9). The control compounds reproduced stable binding poses with binding affinities of –8.3 and –9.0 kcal/mol, RMSD = 0, and consistent interaction patterns (6 H-bonds with CK2α, 4 hydrophobic bonds with TNBC receptor), confirming reliability of the docking protocol. Comparative analysis Table 8 and S1 Table shows that our ligands (L01–L04) exhibit binding affinities comparable to or stronger than the redocked controls. For instance, L02 bound more strongly to 5HA9 (–10.2 kcal/mol) than the control (–9.0 kcal/mol), while L03 achieved –9.1 kcal/mol against CK2α, surpassing the control value (–8.3 kcal/mol). Reference drugs (Doxorubicin, Epirubicin, Docetaxel, Paclitaxel, Olaparib) were also docked for benchmarking. Notably, Olaparib demonstrated the highest affinities (–11.0 and –11.4 kcal/mol), validating the docking protocol's ability to distinguish between moderate and strong binders.

In contrast, when docked against the TNBC receptor (PDB ID: 5HA9), L02 emerged as the most potent binder, with a binding energy of −10.2 kcal/mol, followed closely by L01 (−9.8 kcal/mol) and L03 (−9.1 kcal/mol). Surprisingly, L02 achieved this high affinity without any hydrogen bonds, suggesting the dominance of hydrophobic forces in this interaction as well. While L04 formed the highest number of hydrogen bonds (three) with 5HA9, it exhibited the weakest binding affinity (−6.6 kcal/mol), underscoring that hydrogen bonding alone does not ensure strong binding. The spatial orientation, binding pocket compatibility, and entropy contributions likely play significant roles in modulating the total binding energy.

**Table 8. Binding energy against TNBC between PJT-derived ligands and known TNBC drugs.**

| Ligand | Human CK2 alpha kinase (PUB ID 7L1X) | | | TNBC receptor (PUB ID 5HA9) | | |
|---|---|---|---|---|---|---|
| | Binding Affinity (kcal/mol) | No of H Bond | No of Hydrophobic Bond | Binding Affinity (kcal/mol) | No of H Bond | No of Hydrophobic Bond |
| L01 | −8.3 | 0 | 7 | −9.8 | 2 | 6 |
| L02 | −8.4 | 1 | 4 | −10.2 | 0 | 4 |
| L03 | −9.1 | 0 | 6 | −9.1 | 1 | 2 |
| L04 | −8.6 | 0 | 7 | −6.6 | 3 | 3 |
| Doxorubicin | −8.9 | 4 | 6 | −7.6 | 1 | 8 |
| Epirubicin | −9.4 | 3 | 6 | −8.2 | 1 | 7 |
| Docetaxel | −8.3 | 1 | 8 | −6.3 | 4 | 9 |
| Paclitaxel | −9.2 | 6 | 7 | −6.8 | 5 | 6 |
| Olaparib | −11.0 | 1 | 8 | −11.4 | 5 | 7 |

The designed ligands (L01–L04) demonstrated binding affinities comparable to standard drugs across both CK2α kinase (7L1X) and the TNBC receptor (5HA9). Notably, L02 (–10.2 kcal/mol) and L01 (–9.8 kcal/mol) showed strong binding toward the TNBC receptor, close to Olaparib (–11.4 kcal/mol), the best-performing reference. While standard drugs such as Doxorubicin, Epirubicin, Paclitaxel, and Docetaxel formed more hydrogen bonds overall, the ligands compensated with multiple hydrophobic interactions, suggesting stable occupancy of the active sites. Importantly, L03 exhibited balanced affinity across both proteins (–9.1 kcal/mol each), indicating dual-targeting potential. Thus, the novel ligands, particularly L01–L03, display promising inhibitory activity, in some cases approaching or surpassing the reference drugs, with Olaparib remaining the strongest binder.

Overall, both L02 and L03 demonstrated strong, balanced interactions across both protein targets, making them promising candidates for dual-target inhibition strategies, particularly in the context of multi-target cancer therapeutics. L03, with consistent binding energy of −9.1 kcal/mol for both CK2α and the TNBC receptor, could serve as a versatile lead compound. Conversely, L04 displayed selective binding, performing relatively well with CK2α but poorly with 5HA9, which may be exploited if target specificity toward CK2α is therapeutically desired. These results emphasize the importance of a comprehensive interaction profile, including both quantitative (affinity) and qualitative (interaction type) parameters, in evaluating drug-likeness and target specificity.

### 3.3.2. Protein-ligand interaction.

The interaction diagrams for drug-protein complexes, including hydrogen bonding and molecular docking pockets, were generated using BIOVIA Discovery Studio and PyMOL software. The analysis focused on various hydrogen bond interactions, including conventional and non-conventional hydrogen bonds, as well as hydrophobic interactions such as pi-sigma, alkyl, and pi-alkyl interactions. Additionally, the study examined hydrogen bond donors and acceptors, highlighting their crucial role in drug activity. The graphical representation in Fig 5. illustrates the distinct binding interactions and binding energies between the compound and its target protein.

The protein–ligand interaction analysis provides critical insights into how natural compounds such as Pteryxin, Suksdorfin, Visnadine, and Isosamidin interact with TNBC protein structures (PDB IDs: 7L1X and 5HA9). The binding site residues identified suggest that these ligands occupy and stabilize within the active site pocket, potentially blocking the normal substrate access or catalytic function of the protein targets.

For PDB ID: 7L1X, all four ligands consistently interacted with residues VALA-66, ILEA-174, VALA-53, and LYSA-68, with additional stabilization by MFTA-163 in Suksdorfin, Visnadine, and Isosamidin shown in S2 Table. These residues are part of the hydrophobic and polar core within the active pocket, and their repeated interaction across different ligands indicates that this region is essential for protein function. By forming hydrophobic contacts (VALA-53, VALA-66, ILEA-174) and hydrogen-bonding or electrostatic interactions (LYSA-68), the ligands effectively block substrate entry and alter the conformational dynamics of the binding site. This suggests a competitive inhibition mechanism, where ligands mimic or occupy the natural substrate-binding region, thereby preventing downstream signaling or catalytic activity.

For PDB ID: 5HA9, the interactions involve ILEB-147, ASPB-129, and VALB-184. Here, Aspartate (ASPB-129) may serve as a key catalytic residue, often involved in proton transfer or charge stabilization. The consistent interaction of all ligands with ASPB-129 strongly supports a catalytic site blockade mechanism. Meanwhile, hydrophobic stabilization by ILEB-147 and VALB-184 anchors the ligand in position, ensuring tight occupancy of the binding pocket. Notably, Suksdorfin and Pteryxin lack interaction with ILEB-147, suggesting they may bind more peripherally, possibly leading to weaker inhibition compared to Visnadine and Isosamidin S2 Table).

The ligands employ a dual blocking strategy: (i) hydrophobic interactions with Val and Ile residues provide strong binding and stability, while (ii) catalytic site blockade through Asp and Lys residues prevents essential enzymatic activity. Thus, these ligands may exert anticancer activity in TNBC by competitively blocking the active site, leading to loss of function of key oncogenic proteins. Among the tested ligands, Visnadine and Isosamidin appear to provide stronger inhibitory potential due to their consistent interaction with both hydrophobic and catalytic residues across different protein conformations, suggesting they may serve as lead candidates for further drug development.

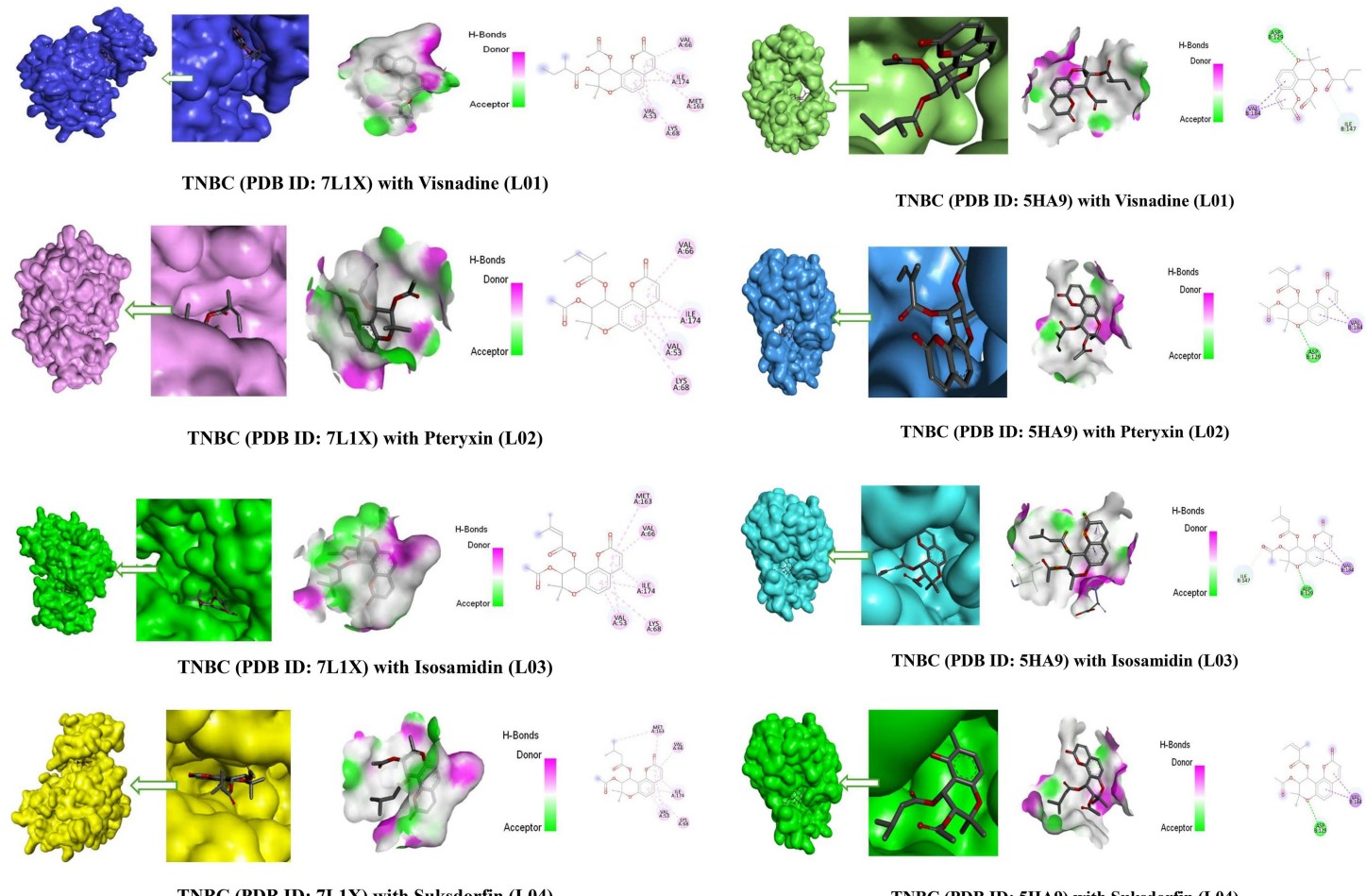

**Fig 5. The results of molecular docking pose interactions between compounds and proteins.**

### 3.4. Molecular dynamics simulation

**3.4.1. Protein RMSD.** The graph titled Protein RMSD (Å) illustrates the Root Mean Square Deviation (RMSD) of the protein structure over the course of a molecular dynamic simulation. The x-axis represents simulation time (in nanoseconds or minutes), while the y-axis indicates RMSD values in Ångströms (Å), reflecting structural deviations from the initial conformation explained in Fig 6.

RMSD fluctuations across the timeline provide insights into the conformational stability of the protein. Lower RMSD values (e.g., ~1.5 Å) suggest structural stability, whereas higher values (e.g., ~3.4 Å) indicate significant conformational changes. Distinct lines or colors such as blue for chain A, red for chain B, and green for a particular domain highlight the dynamic behavior of specific regions.

Key time points, such as 10 nsec (1.8 Å), 20 nsec (2.5 Å), and 30 nsec (3.2 Å), demonstrate gradual or abrupt changes in RMSD, possibly pointing to structural rearrangements. Overall, this graph provides valuable information about the protein's flexibility and stability throughout the simulation period.

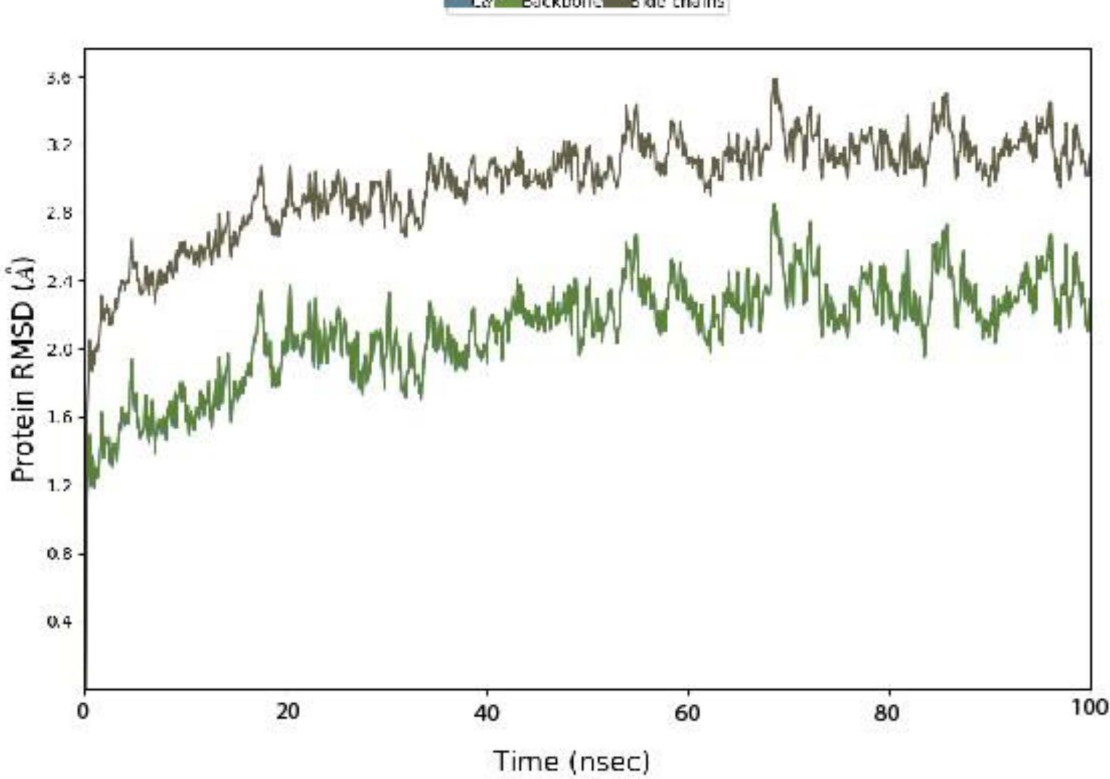

**Fig 6. Protein RMSD (Å) with respect of time frame.**

**3.4.2. Ligand RMSD.** The graph titled Ligand RMSD likely illustrates the Root Mean Square Deviation (RMSD) of a ligand over time, with the x-axis representing time in nanoseconds (nsec) and the y-axis showing RMSD values in Ångströms (Å) shown in Fig 7.

The graph may include two lines: one where the ligand is fitted on the protein (Lig fit on Prot) and another where the ligand is fitted on itself (Lig fit on Lig), possibly represented by different colors such as blue and red, respectively. The RMSD values fluctuate over time, starting around 0.8 Å and gradually increasing to values like from 2.4 Å to 3.6 Å, and peaking at higher values such as 3.4 Å or 3.6 Å, indicating conformational changes or instability in the ligand's binding. Specific time points, such as 10 nsec (1.2 Å), 50 nsec (3.2 Å), and 100 nsec (3.6 Å), highlight how the ligand's position or structure evolves. Higher RMSD values suggest greater deviations, possibly due to ligand flexibility or binding site dynamics, while lower values indicate stability. This graph helps analyze the ligand's behavior and interaction with the protein during the simulation.

**3.4.3. Protein RMSF with x-axis representing the residue index.** The graph titled Protein RMSF (Root Mean Square Fluctuation) visualizes the flexibility of different parts of a protein, with the x-axis representing the residue index (ranging from 0 to 700) and the y-axis showing RMSF values in Ångströms (Å) added in Fig 8.

The graph likely includes three lines or regions: C-alphas (possibly in blue), Backbone (possibly in red), and Side chains (possibly in green), each representing the fluctuations of these specific protein components. Higher RMSF values indicate greater flexibility, while lower values suggest rigidity. For example, residues 50–100 might show RMSF values around 1.5 Å for C-alphas, 2.0 Å for the backbone, and 2.5 Å for side chains, indicating that side chains are more flexible. Peaks at specific residues, such as residue 200 (3.0 Å) or residue 400 (4.5 Å), highlight regions of high mobility, possibly

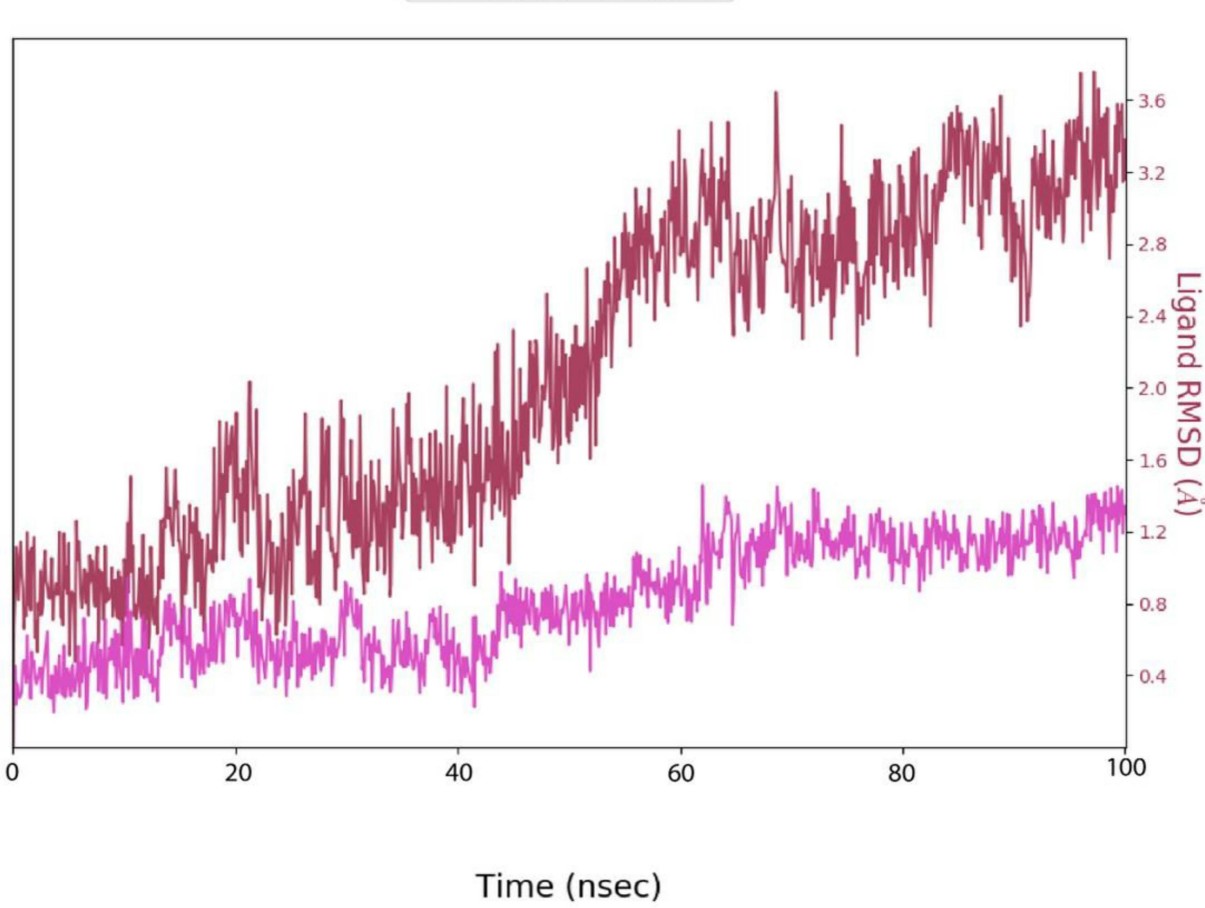

**Fig 7. Ligand RMSD (Å) with respect of time frame.**

due to loops or unstructured regions. This graph helps identify which parts of the protein are stable or dynamic, providing insights into its functional and structural behavior.

**3.4.4. Protein-ligand contacts.** The graph titled P-L Contacts (Protein-Ligand Contacts) illustrates the interactions between a protein and a ligand over time, with the x-axis representing time in nanoseconds (nsec) and the y-axis showing the fraction of interactions conveyed in Fig 9.

The graph likely includes different types of interactions, such as H-bonds (possibly in blue), hydrophobic interactions (possibly in green), ionic interactions (possibly in red), and water bridges (possibly in purple). Each interaction type is represented by a colored line or bar, indicating the frequency or strength of the interaction over time. For example, H-bonds might show a fraction of 0.5 at 20 nsec, increasing to 0.7 at 50 nsec, while hydrophobic interactions might remain steady around 0.3. Specific residues like GLN_198, ASP_102, and ARG_204 are highlighted for their frequent contact with the ligand, with frame counts indicating how often these interactions occur. The histogram provides detailed residue-specific contact information, such as TVR_49 interacting with multiple residues like GLN_198 and ASP_102. This graph helps analyze the stability and nature of protein-ligand interactions, providing insights into binding mechanisms and key residues involved.

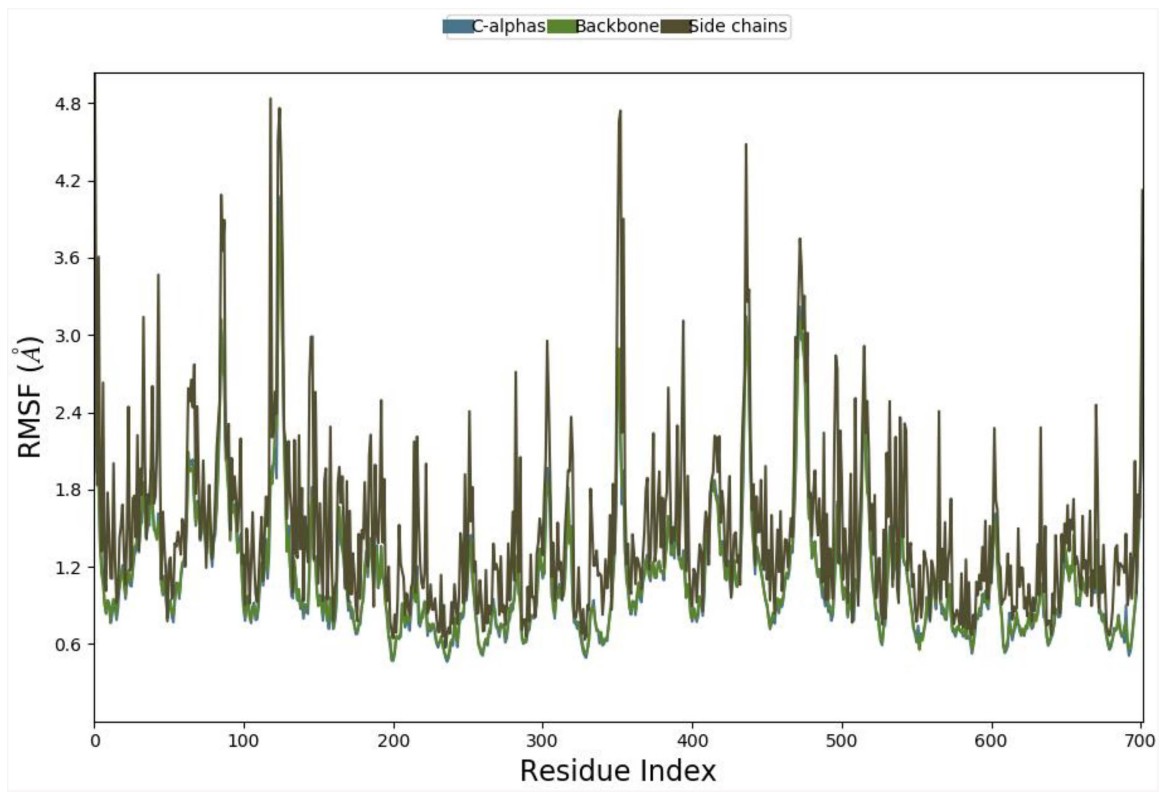

**Fig 8. Protein RMSF with x-axis representing the residue index.**

Key hydrogen bonds were observed with residues ASP102, ASP105, ASN106, SER203, and ARG204, with interaction fractions ranging between 0.2–0.6, indicating moderate-to-strong stability. Notably, ARG217 showed the most persistent hydrogen bonding (>1.0 interaction fraction), highlighting its dominant role in stabilizing the complex. Water-mediated bridges, frequently involving TYR228, TYR235, and TYR246, further contributed to dynamic stabilization, while hydrophobic interactions with ILE207, LEU216, ILE218, and LEU234 (fractions ~0.5–0.8) provided additional anchoring support. Although ionic interactions were less frequent, they contributed transient reinforcement to the binding process.

The contact profile and structural compactness of the complex further underscored its stability. The number of total interactions remained consistently within 6–10 throughout the simulation, with occasional peaks of ~12, while consistent contacts with residues such as ARG217, ASP105, and TYR235 minimized dissociation risks. The radius of gyration (Rg) remained stable across the trajectory, confirming that the protein–ligand complex retained a compact conformation with minimal structural drifts. Collectively, these findings indicate energetically favorable, persistent interactions, dominated by hydrogen bonding with ARG217 and hydrophobic contacts, thereby supporting the strong and stable binding affinity of the ligand.

The graph depicts ligand flexibility (RMSF), showing differences in mobility depending on alignment, with peaks indicating highly flexible regions shown in S3 Fig. Again, the L-Properties graph shows how a ligand's flexibility, volume, torsion, solvent-accessible surface area, and polar surface area change over time, with no intramolecular hydrogen bonds detected from S4 Fig. Additionally, the B-factor graph shows protein residue flexibility from S5 Fig, with higher values indicating more mobile regions—peaks highlight highly dynamic areas, while lower values mark stable, rigid regions. Next, the PSS graph from S6 Fig shows how a protein's secondary structures—helices, strands, and loops—change over time,

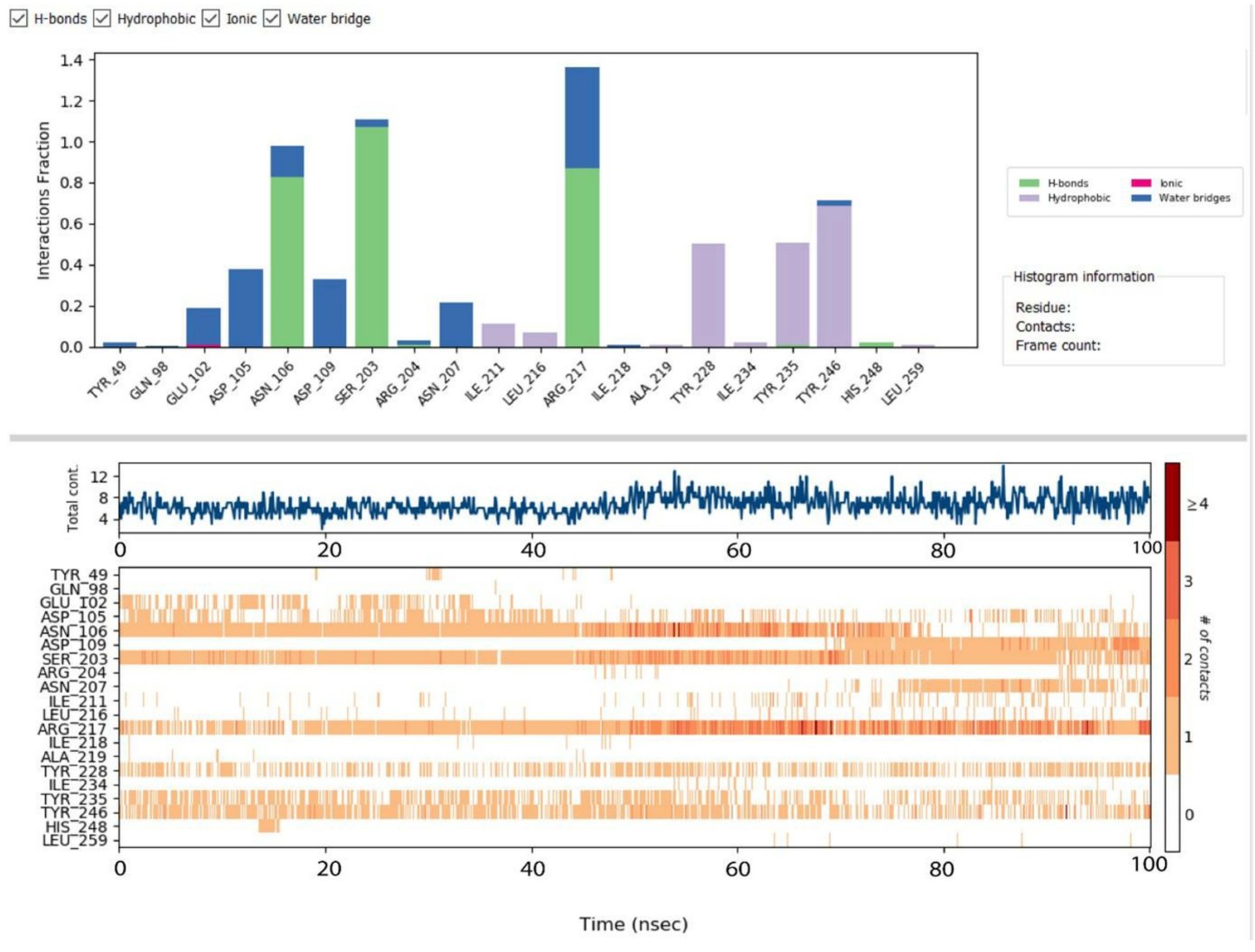

**Fig 9. Protein-Ligand Contacts.**

highlighting stable regions and structural transitions in the protein-ligand complex. The ligand forms specific interactions with the target from S7 Fig, involving key residues that stabilize the complex: ASN-106 (55%), SER-203 (86%), and ARG-217 (87%) through charged or polar interactions, and TYR-228, TYR-235, and TYR-246 through hydrophobic contacts.

The S8 Fig illustrates the torsional flexibility of a ligand through torsion angle plots and corresponding circular histograms for different rotatable bonds, showing the distribution of sampled dihedral angles during molecular docking or molecular dynamics simulations. Peaks in the histograms indicate energetically favorable torsion angles, with blue- and orange-highlighted bonds showing concentrated peaks and restricted rotational flexibility, while red- or purple-highlighted bonds display broader distributions and greater conformational freedom. Overlaid energy curves reveal torsional energy barriers, with sharp minima indicating rigidity and shallow profiles indicating flexibility, which influences how the ligand fits into the protein binding pocket. This flexibility affects binding, as flexible bonds can adapt to pocket geometry to enhance affinity and specificity, whereas rigid regions maintain structural integrity and support stable interactions like hydrogen

bonds, hydrophobic contacts, and π-stacking. Correlating torsion angles with docking results shows that flexible torsions facilitate alignment of functional groups with key residues (e.g., ASN-106, SER-203, ARG-217, or TYR residues), and color-coded torsions in the 2D ligand structure link specific regions to adaptable or constrained binding. Understanding torsional flexibility informs rational ligand design, as highly flexible bonds can be chemically modified to optimize movement and binding properties, while rigid segments are preserved to maintain essential pharmacophoric features.

## 4. Conclusion

This comprehensive computational study highlights the therapeutic promise of dihydropyranocoumarins derivatives (L01–L04) as targeted agents against triple-negative breast cancer (TNBC). Among these, L02 emerges as the lead candidate, exhibiting the highest chemical reactivity and flexibility suited for dynamic protein environments, along with the strongest binding affinity to the TNBC receptor (5HA9) and favorable interaction with CK2α. Its binding is driven mainly by moderate hydrophobic contacts and minimal hydrogen bonding, indicating an optimal fit within hydrophobic pockets. L03 shows strong and consistent binding to both targets, acting as a potential dual inhibitor, with balanced stability and reactivity. L01 and L04 serve as useful middle-ground candidates, with L01 showing balanced efficacy and slightly better solubility, while L04 demonstrates the strongest electron-accepting properties. As a result, poor binding affinity likely due to conformational strain or desolvation effects despite high hydrogen bonding. Finally, L02 was designated as the lead candidate because it showed the strongest binding affinity against the TNBC receptor (–10.2 kcal/mol), approaching the performance of the standard Olaparib. On the other hand, L03 was highlighted for its balanced binding profile, displaying comparable affinities against both CK2α kinase and the TNBC receptor (–9.1 kcal/mol each), suggesting dual-targeting potential. Thus, while L02 is prioritized as the primary lead due to its potency, L03 represents a promising multi-target scaffold that could be further optimized for broader therapeutic applications.

All four compounds display favorable oral absorption and minimal metabolic liabilities, with negligible blood–brain barrier permeability and CYP450 inhibition, suggesting safety and suitability for non-CNS therapeutic applications. However, poor aqueous solubility across the series remains a challenge, which may be addressed via advanced formulation strategies like salt formation, nano formulations, or prodrugs. Biological activity predictions further support L02's superior anticancer and antimicrobial potential, with L03 also showing strong antineoplastic and antibacterial activity. L01 and L04 exhibit broad-spectrum effects but with relatively lower efficacy. Moderate antiviral activity across all ligands indicates a likely supportive rather than primary therapeutic role. Together, these results underscore the importance of evaluating holistic interaction profiles including hydrophobic contacts, electronic properties, and solubility in rational drug design. L02 and L03 stand out as the most promising candidates for further preclinical development in oncology and infectious disease, offering a strong foundation for the design of effective, targeted anticancer therapies [31]. The findings contribute valuable insights toward identifying nature-derived therapeutics for TNBC and set the stage for subsequent in vitro and in vivo validations although there are added some result in S3 Table.

However, it is important to note that these findings are predictive and based solely on in silico analyses. Therefore, further experimental validation such as in vitro cytotoxicity testing, kinase inhibition assays, and in vivo studies are essential to confirm the therapeutic potential of these compounds.

## Supporting information

**S1 Table. Data for redocking procedure by control.**
(DOCX)

**S2 Table. Protein ligand interaction data.**
(DOCX)

**S3 Table. Growth and serum parameters of in vivo studies.**
(DOCX)

**S1 Fig. Ramachandran plots of (A) PARP1 (PDB ID: 5HA9) and (B) CK2α (PDB ID: 7L1X) generated using PROCHECK.**
(DOCX)

**S2 Fig. Bioavailability radar plots for PJT-derived ligands (L01–L04) and known TNBC drugs (Paclitaxel, Docetaxel, Doxorubicin, Epirubicin).**
(DOCX)

**S3 Fig. The graph titled "L-RMSF" (Ligand Root Mean Square Fluctuation) illustrates the flexibility of a ligand, with the x-axis representing atom indices (e.g., Aton, Bton, Cton, Dton) and the y-axis showing RMSF values in Ångströms (Å).**
(DOCX)

**S4 Fig. Radius of gyration (Rg) of WT, mutations, Solvent-accessible surface area (SASA).**
(DOCX)

**S5 Fig. The graph titled "P-RMSF B factor" visualizes the Root Mean Square Fluctuation (RMSF) and B-factor (temperature factor) of a protein, with the x-axis representing the residue index (ranging from 0 to 700) and the y-axis showing the B-factor values.**
(DOCX)

**S6 Fig. Protein secondary structure elements (SSE) of protein- ligand complexes.**
(DOCX)

**S7 Fig. Molecular interaction between the ligand and target.**
(DOCX)

**S8 Fig. Torsional flexibility In Molecular Docking Studies protein- ligand complexes.**
(DOCX)

## Acknowledgments

Authors are expressed their thankful to the IUBAT Innovation and Entrepreneurship Center (IIEC), IUBAT—International University of Business Agriculture and Technology, 4-Embankment Drive Road, Sector 10 Uttara Model Town, Dhaka 1230, Bangladesh for their technical supports to carry on this research progress.

## Author contributions

**Conceptualization:** Abu Yousuf Hossin, Md. Naziur Rahman, Md Mahabub Hasan, Ansarul Karim, Shyikh Ahmed Alif, Naziat Nayel Arshi, Shammi Jahan.

**Data curation:** Abu Yousuf Hossin, Md. Naziur Rahman, Md Mahabub Hasan, Ansarul Karim, Shyikh Ahmed Alif, Naziat Nayel Arshi, Shammi Jahan.

**Formal analysis:** Abu Yousuf Hossin, Md. Naziur Rahman, Ansarul Karim, Naziat Nayel Arshi, Shammi Jahan.

**Funding acquisition:** Ajoy Kumer.

**Investigation:** Abu Yousuf Hossin.

**Methodology:** Abu Yousuf Hossin, Md. Naziur Rahman, Md Mahabub Hasan.

**Project administration:** Ajoy Kumer.

**Resources:** Abu Yousuf Hossin, Md. Naziur Rahman, Md Mahabub Hasan, Ansarul Karim, Shyikh Ahmed Alif, Naziat Nayel Arshi.

**Software:** Naziat Nayel Arshi.

**Supervision:** Ajoy Kumer.

**Visualization:** Md. Naziur Rahman, Md Mahabub Hasan, Ansarul Karim, Shyikh Ahmed Alif, Shammi Jahan.

**Writing – original draft:** Abu Yousuf Hossin, Shammi Jahan, Ajoy Kumer.

**Writing – review & editing:** Ajoy Kumer.

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
