## [Decision Letter · Decision Letter 0]

26 Jul 2025

Dear Dr. Kumer,

Thank you for submitting your manuscript to PLOS ONE. After careful consideration, we feel that it has merit but does not fully meet PLOS ONE’s publication criteria as it currently stands. Therefore, we invite you to submit a revised version of the manuscript that addresses the points raised during the review process.

**ACADEMIC EDITOR:**

We look forward to receiving your revised manuscript.

Kind regards,

Yusuf Oloruntoyin Ayipo, Ph.D

Academic Editor

PLOS ONE

Journal Requirements:

3. We note that your Data Availability Statement is currently as follows: No

4. Please ensure that you refer to Figure 5, 6, and 7 in your text as, if accepted, production will need this reference to link the reader to the figures.

5. Please include your tables as part of your main manuscript and remove the individual files. Please note that supplementary tables (should remain/ be uploaded) as separate "supporting information" files"

Additional Editor Comments:

The submission reflects scientific relevance. However, some fundamental issues limit its quality for publication in the current form. For instance, the authors need to justify the significance of the study, relate it to the literature and identify the gap in the existing knowledge that this aims to satisfy, and ensure an adequate validation of the theoretical analysis. Again, what are the limitations of this study and how can the authors recommend future research on the study. Moreover, some concerns have been raised by the reviewers affecting certain sections of the study. Kindly pay a thorough attention to these and address them critically before resubmission.

Reviewers' comments:

Reviewer's Responses to Questions

**Comments to the Author**

1. Is the manuscript technically sound, and do the data support the conclusions?

Reviewer #1: Yes

Reviewer #2: Yes

Reviewer #3: Partly

Reviewer #4: Yes

2. Has the statistical analysis been performed appropriately and rigorously?

Reviewer #1: Yes

Reviewer #2: Yes

Reviewer #3: N/A

Reviewer #4: Yes

3. Have the authors made all data underlying the findings in their manuscript fully available?

Reviewer #1: Yes

Reviewer #2: Yes

Reviewer #3: No

Reviewer #4: Yes

4. Is the manuscript presented in an intelligible fashion and written in standard English?

Reviewer #1: Yes

Reviewer #2: Yes

Reviewer #3: Yes

Reviewer #4: Yes

Reviewer #1: the manuscript is well written and the diagrams ar well made and well explained, the concepts are explained thourougly with providing all the background data that is being used which is good for the person who wants to replicate the research or needs background data.

the tables are also well explained in the passage.

Reviewer #2: Evaluation of Dihydropyranocoumarins as Potent Inhibitors Against Triple-Negative Breast Cancer: An Integrated DFT, in silico approaches, Docking and MD Study

1. Overview

(1.I). The manuscript investigates potential new treatment strategies for TNBC, which is a clinically challenging subtype of cancer with unmet therapeutic needs. Targeting it with natural product derivatives is a promising strategy.

(1.II). The manuscript seeks to fill a critical knowledge gap on a clinically important topic, but will benefit from a revision across the areas described below.

2. Title

(2.I). Avoid vague terms like “in silico approaches.” Instead, specify the techniques used. Consider rephrasing for clarity and precision to: “Integrated Quantum Chemical and Molecular Modeling Study of Dihydropyranocoumarins as Potential Inhibitors of Triple-Negative Breast Cancer”

3. Abstract

(3.I). Docking against 5HA9 and 7L1X is good, but the biological relevance of these targets to TNBC must be justified in the full text.

(3.II). The binding energy of −9.1 kcal/mol is promising, but comparison with known TNBC inhibitors would strengthen the claim.

(3.III). The claim of “high probabilities for antineoplastic activity” needs quantitative backing.

(3.IV). The lack of experimental validation is a limitation. Therefore, consider proposing in vitro assays, such as MTT, apoptosis, and migration, as future directions.

(3.V). To improve flow, clarity, and readability, consider simplifying the phrase “...corroborated by MEP maps and charge distribution analyses” and making the phrase “Dynamic compatibility under physiological conditions” more specific, like “stable RMSD and compact protein-ligand complex.”

4. Introduction

(4.I). The introduction is dense and occasionally repetitive. For example, the paragraph beginning with “Therefore, TNBC has fewer treatment options…” repeats earlier points. Consider merging or trimming overlapping content to improve flow.

(4.II). The sentence “To the best of our knowledge, we are the first group to report on the anti-cancer properties of coumarins” is factually incorrect since coumarins have been widely studied for anticancer activity. Maybe you should rather say: “To the best of our knowledge, this is the first comprehensive in silico evaluation of dihydropyranocoumarins from Peucedanum japonicum Thunb as potential TNBC therapeutics.”

(4.III). Some references are incomplete or unclear. For example, “World Cancer Day 2021” is not a proper citation. Also, “Bcrf, 2024” should be replaced with a peer-reviewed source or a properly formatted web citation. Ensure that all references are either peer-reviewed or from authoritative databases.

(4.IV). The introduction lacks a clear rationale for the selected TNBC protein targets (e.g., 5HA9 and 7L1X). Consider adding a brief explanation of their biological roles in TNBC (e.g., kinases, transcription factors, etc.).

(4.V). Several grammatical issues need correction. For example, “This is because TNBC lacks the expression…” can be better phrased as “This is due to the absence of ER, PR, and HER2 expression, rendering hormone and HER2-targeted therapies ineffective.” Also, the phrase “So, the perspective of this research…” can be revised as “Therefore, this study aims to identify…”

5. Materials and Methods

(5.I). On Ligand Preparation and DFT Optimization, there is inconsistency in the functional used as both GGA-PBE and B3LYP are mentioned. Clearly state which functional was used for each calculation.

(5.II). On Ligand Preparation and DFT Optimization, there is mention of basis set superposition error (BSSE) correction or solvent model (e.g., COSMO, PCM), which could affect accuracy. Justify the absence of solvent effects or include them if relevant to biological conditions.

(5.III). On PASS Prediction, the use of Pa > Pi threshold is appropriate. However, include confidence intervals or Pa values for antineoplastic activity.

(5.IV). On PASS Prediction, the biological relevance of the predicted activities is not critically discussed. Consider discussing the limitations of PASS predictions, such as false positives and lack of target specificity

(5.V). The “On Lipinski Rule and Pharmacokinetics” section is descriptive but lacks comparative analysis. For example, how do these values compare to known TNBC drugs?. Consider including a summary table comparing all compounds across Lipinski parameters. Also, discuss bioavailability radar or BOILED-Egg plots if available.

(5.VI). On ADMET Profiling, the Use of pkCSM and ADMETsar is appropriate. However, hepatotoxicity is predicted but dismissed based on traditional use. This is scientifically weak. Consider acknowledging the computational prediction of hepatotoxicity and suggesting in vitro validation. Also, there is no mention of toxicity class, LD50 categories, or organ-specific toxicity. Consider including toxicity class (I–V) and toxicity endpoints such as hERG inhibition and skin sensitization.

(5.VII). On Protein Preparation, there was no mention of validation of protein structures using, for example, Ramachandran plot, PROCHECK, Et cetera.

(5.VIII). On Protein Preparation, there was no justification for choosing PDB IDs 5HA9 and 7L1X. On Protein Preparation, consider including target selection rationale (e.g., CK2α relevance in TNBC).

(5.IX). On Protein Preparation, report protein quality metrics (e.g., resolution, R-free, % favored residues).

(5.X). On Molecular Docking, there was no validation of the docking protocol (e.g., redocking native ligand, RMSD < 2 Å. Consider including docking validation and binding site analysis.

(5.XI). On molecular Docking, there was no comparison to known inhibitors or positive controls. Consider reporting interacting residues and types of interactions such as H-bonds, π–π, etc.

(5.XII). On Molecular Dynamics (MD) Simulations, there was no mention of MM-PBSA/MM-GBSA binding free energy calculations. Include MM-PBSA/MM-GBSA to estimate binding free energies. Also, report temperature, pressure control, and equilibration steps.

(5.XIII). On Molecular Dynamics (MD) Simulations, there were no replicates or error analysis. Consider triplicate runs for statistical robustness.

6. Result and Discussions

(6.I). On DFT and Molecular Descriptors, the functional inconsistency (B3LYP vs. PBE) noted in the Methods is not addressed here. Clarify which DFT functional was used for each descriptor.

(6.II). The biological relevance of descriptors like softness and hardness is not clearly linked to TNBC inhibition. Discuss how these descriptors might influence ligand–protein interactions or cell permeability.

(6.III). On Electrostatic Potential Maps, highlight how negative potential regions might favor interactions with positively charged residues in the TNBC targets.

(6.IV). On PASS Prediction, the Pa values for antineoplastic activity are not contextualized. For example, it is unclear what is considered a strong prediction. Consider including a threshold interpretation (e.g., Pa > 0.7 = high confidence. Also, discuss false positive risks and the need for experimental validation.

(6.V). On Lipinski and ADMET Profiling, the hepatotoxicity prediction is downplayed based on traditional use, which is not scientifically sufficient. Acknowledge the computational prediction of hepatotoxicity and recommend in vitro liver toxicity assays.

(6.VI). There was no comparison to known TNBC drugs such as doxorubicin, olaparib, Et cetera. Consider comparing ADMET profiles to reference drugs to contextualize the findings.

(6.VII). On Molecular Docking, there was no validation of docking protocol. For example, in redocking native ligands, RMSD < 2 Å. Include docking validation and reference ligand comparisons.

(6.VII). On Molecular Docking, there was no comparison to known inhibitors of CK2α or TNBC targets. Discuss binding site residues and their known roles in TNBC biology.

(6.VIII). On Molecular Dynamics (MD) Simulations, there was no MM-PBSA/MM-GBSA binding free energy calculations reported. Include binding free energy estimates to strengthen conclusions.

(6.IX) There are no replicates or error bars shown. Discuss conformational stability in terms of RMSD plateaus and contact persistence.

(6.X). On Interpretation and Biological Relevance, the discussion lacks depth in terms of mechanistic hypotheses. For example, it is unclear how these compounds might inhibit TNBC progression. Consider proposing mechanisms of action based on docking site (e.g., kinase inhibition, DNA intercalation).

(6.XI). There was no mention of potential off-target effects or selectivity. Consider recommending experimental assays (e.g., MTT, apoptosis, kinase inhibition) to validate predictions.

7. Conclusion

(7.I). The conclusion implies therapeutic efficacy based solely on in silico data, which can be misleading. Consider adding a disclaimer that these findings are predictive and require experimental validation, such as via in vitro cytotoxicity, kinase inhibition assays, or in vivo models.

(7.II). The conclusion does not mention the biological targets (e.g., CK2α, 5HA9) or their relevance to TNBC. Briefly restate the mechanistic hypothesis or target pathway to reinforce the biological rationale.

(7.III). There is no comparison to existing TNBC therapies or known inhibitors. Mention how the binding affinities or ADMET profiles compare to standard-of-care agents (e.g., doxorubicin, olaparib) to contextualize the novelty and potential impact.

(7.IV). The conclusion states that L02 is the “lead candidate” but also emphasizes L03’s dual-target potential, which might confuse readers. Clarify the distinct advantages of each compound.

(7.V). Some sentences are overly long and dense. Consider breaking up complex sentences for clarity and impact. Also, avoid redundancy. For example, the phrases “strongest electron-accepting properties” and “high hydrogen bonding” are repeated without added value.

8. Final Recommendations

(8.I). The manuscript seeks to fill a critical knowledge gap on a clinically important topic, but will benefit from a revision across the areas described above.

Reviewer #3: Major Concerns

- Introduction

The second to the last sentence in the introduction section; "This study was aimed to characterize the TNBC properties of dihydropyranocoumarins in in-silico studies" should be reframed. The sentence is not logically correct.

The last sentence in the introduction section; "To the best of our knowledge, we are the first group to report on the anti-cancer properties of coumarins" should be removed or rephrased. You are not the first group to report on the anti-cancer properties of coumarins. A simple google search using the prompt "anti-cancer properties of coumarins" would show you numerous papers that have broadly addressed this. You want to take out the statement or re-phrase it to be more specific to PJT, anti-TNBC (instead of just saying anti-cancer), dihydropyranocoumarins (instead of saying coumarins). You can also make it specific to the computational studies being carried out.

- Computational Chemistry & DFT

The manuscript inconsistently refers to using both GGA-PBE and B3LYP for DFT. Please clarify which functional was ultimately used for geometry optimization and why.

The basis set used is referred to as DNP, but also mentions 6-31G** in another section. Clarify this contradiction.

No validation or comparison with experimental geometries or energies was done. Consider referencing prior studies validating DFT-predicted geometries for coumarins or similar compounds.

- Docking Methodology

The binding site(s) were not explicitly validated (e.g., with control ligands, literature-reported residues, or co-crystallized ligands). This could lead to docking in non-biologically relevant pockets.

Docking results would benefit from statistical or comparative validation e.g., a known TNBC inhibitor redocked as a positive control.

- Molecular Dynamics (MD) Simulations

Although 100 ns MD is adequate, RMSD values for the ligand rise to 10.6 Å, which indicates substantial instability or unbinding. This contradicts the narrative of strong binding. Please clarify and explain this anomaly.

The MD section lacks quantitative summaries like radius of gyration, hydrogen bond occupancy, or total interaction energy. Kindly include details of these

Protein preparation for MD lacks critical details (e.g., protonation states, force field for ligands, water model used). Kindly include details of these

- Tables

All the tables (Tables 1-9 are missing). Only the table titles are in the manuscript. Kindly put the tables in the manuscript.

- Results and Discussion

These sections should be separate and not together. Discussion section is one where you interpret and conceptualize your results (include citations to support your interpretation). Here, you want to explain the implications of your findings, compare with existing literature (include citations to support), highlight unexpected results, talk about limitations, future directions etc.

- Minor Issues

Typos (e.g., “silico” instead of “in silico,” “radish brown” instead of “reddish brown”).

- Recommendations for the Authors

Clarify the DFT functional/basis set and ensure consistency.

Improve docking validation by including known ligands, active-site residue justification, and clearer grid setup.

Address the ligand instability in MD, was there unbinding, or is it an artifact of simulation setup?

Consider reducing redundancy in sections and ensuring grammar and terminology are polished.

Reviewer #4: I have examined the publication entitled " Evaluation of Dihydropyranocoumarins as Potent Inhibitors Against Triple-Negative Breast Cancer: An Integrated DFT, in silico approaches, Docking and MD Study. This publication examines the combined anti-cancer efficacy of ONC201 and MAPK pathway inhibitors (trametinib, ulixertinib, and AMG-510) in triple-negative breast cancer (TNBC). The research indicates that ONC201 triggers apoptosis in TNBC cells and, when used in conjunction with MAPK inhibitors, markedly increases cell death and suppresses tumor growth both in vitro and in vivo. The authors propose that dual targeting of ERK signaling, and ClpP/ATF4-mediated apoptosis is a viable treatment approach for TNBC. However, I have identified several areas needing modification before the manuscript may be recommended for publication.

Strengths:

Scope and Relevance:

This research addresses a significant unmet requirement in TNBC, a subtype devoid of effective targeted treatments.

It also evaluates ONC201, a small drug presently undergoing clinical review, in a strategic combination therapy with MAPK pathway inhibitors.

The notion of synthetic lethality via mitochondrial ClpP activation and ERK pathway inhibition is innovative and clinically significant.

Methodology:

The authors utilized appropriate assays: Cell viability, Western blot, apoptosis, qRT-PCR, colony formation, and animal models. Several inhibitors (trametinib, ulixertinib, AMG-510) were utilized to analyze the components of the MAPK pathway involved in synergy.

Presentation:

The manuscript presents a coherent and compelling rationale for targeting TNBC using a dual strategy involving ONC201 and MAPK inhibitors. Also, each section (Introduction, Methods, Results, Discussion) flows logically and supports the central hypothesis.

Suggestions:

Numerous figure legends and result descriptions require more detail, such as the specification of doses and time points utilized in investigations.

Certain acronyms (e.g., ISR, ClpP) are employed without prior elucidation.

Several statements in the Discussion section are excessively lengthy or conjectural and might benefit explanation.

In the Results, explicitly differentiate between in vitro and in vivo observations.

Limitations:

The Synergism between ONC201 and MAPK inhibitors is described but lacks formal synergy quantification.

Only a single xenograft model (MDA-MB-231) is used and no histopathological or IHC analysis of tumors to confirm pathway inhibition or apoptosis markers in vivo.

Some bar graphs lack error bars or significance indicators.

**Do you want your identity to be public for this peer review?** For information about this choice, including consent withdrawal, please see our Privacy Policy

Reviewer #1: **Yes: ** Shelly Saima Yaqub

Reviewer #2: No

Reviewer #3: No

Reviewer #4: No

---

## [Author Response · Author response to Decision Letter 1]

23 Sep 2025

PONE-D-25-33604R1

Evaluation of Dihydropyranocoumarins as Potent Inhibitors Against Triple-Negative Breast Cancer: An Integrated of In silico, Quantum & Molecular Modeling Approaches

Dear Dr. Kumer,

We've checked your submission and before we can proceed, we need you to address the following issues:

1. Please provide a complete Data Availability Statement in the submission form, ensuring you include all necessary access information or a reason for why you are unable to make your data freely accessible. If your research concerns only data provided within your submission, please write "All data are in the manuscript and/or supporting information files" as your Data Availability Statement.

Answer: It has revised and added.

2. Please ensure that you refer to Figure 5, 6, and 7 in your text as, if accepted, production will need this reference to link the reader to the figures.

Answer: It has revised and added.

3. Please include a copy of Table 1-8 which you refer to in your manuscript file.

Answer: It has revised and added.

4. Please upload a Response to Reviewers letter which should include a point-by-point response to each of the points made by the Editor and / or Reviewers. (This should be uploaded as a 'Response to Reviewers' file type.) Please follow this link for more information: http://blogs.PLOS.org/everyone/2011/05/10/how-to-submit-your-revised-manuscript/

Answer: It has revised and added.

5. We notice that your revision was submitted on [Sep 11 2025], but the manuscript file in your submission's file inventory was uploaded on [Jul 1 2025]. Please upload the latest version of your revised manuscript as the main article file, with the item type 'Manuscript,' ensuring that it does not contain any tracked changes or highlighting. This will be used in the production process if your manuscript is accepted.

Answer: It has revised and added.

Response to the Editor and Reviewers

Manuscript ID: PONE-D-25-33604

Title: Evaluation of Dihydropyranocoumarins as Potent Inhibitors Against Triple-Negative Breast Cancer: An Integrated DFT, in silico approaches, Docking and MD Study

We sincerely thank the Academic Editor and all reviewers for their valuable time and insightful comments. We have revised our manuscript thoroughly and addressed all comments point by point below. All changes have been incorporated into the revised version, with tracked changes indicated. We believe these revisions have significantly improved the quality and clarity of our manuscript.

Academic Editor Comment

Q1. The submission reflects scientific relevance. However, some fundamental issues limit its quality for publication in the current form. For instance, the authors need to justify the significance of the study, relate it to the literature and identify the gap in the existing knowledge that this aims to satisfy, and ensure an adequate validation of the theoretical analysis. Again, what are the limitations of this study and how can the authors recommend future research on the study. Moreover, some concerns have been raised by the reviewers affecting certain sections of the study. Kindly pay a thorough attention to these and address them critically before resubmission.

Response

We thank the Academic Editor for highlighting the need to more clearly justify the significance of our study, identify the literature gap, articulate the study’s contribution, and provide adequate validation and future research direction.

In response, we have thoroughly revised the Introduction section to explicitly outline the unmet clinical challenges of TNBC, the limitations of current therapeutic strategies, and the promising role of natural products particularly dihydropyranocoumarins from Peucedanum japonicum Thunb. in anticancer drug discovery. We emphasized that despite the pharmacological interest in this plant, no prior research has systematically evaluated its dihydropyranocoumarin derivatives against TNBC targets using a comprehensive in silico strategy. This allowed us to clearly position our work within the existing body of literature and define the knowledge gap our study aims to address.

Furthermore, we have explicitly stated our research contribution, which involves the first integrated computational pipeline using DFT calculations, molecular docking, ADMET screening, and 100-ns MD simulations to assess the anti-TNBC potential of these compounds against two clinically relevant TNBC targets (PARP1 and CK2α). This approach provides mechanistic insight into their therapeutic relevance and offers a rational basis for further preclinical development.

To ensure balanced interpretation, we have added a disclaimer in the conclusion section clarifying that our findings are predictive and require experimental validation, such as in vitro cytotoxicity assays and in vivo studies. Additionally, we briefly outlined the limitations of our in-silico approach and proposed specific directions for future research, including experimental validation and formulation strategies to address solubility issues.

We have carefully considered all the concerns raised by the reviewers and addressed each point thoroughly in the revised manuscript. We critically reviewed the comments to improve the quality and clarity of our work before resubmission. We believe these revisions have significantly strengthened the paper and hope it now meets the standards for publication.

Reviewer #1

Q2. The manuscript is well written and the diagrams are well made and well explained, the concepts are explained thoroughly with providing all the background data that is being used which is good for the person who wants to replicate the research or needs background data.

the tables are also well explained in the passage.

Response

Thank you for the positive feedback. We have preserved the clarity and added further improvement in consistency and figure references where applicable.

Reviewer #2

Comment

1. Overview

Q3 (1.I). The manuscript investigates potential new treatment strategies for TNBC, which is a clinically challenging subtype of cancer with unmet therapeutic needs. Targeting it with natural product derivatives is a promising strategy.

Response

First of all, TNBC lacks effective molecular-targeted therapies, making it a priority for novel computational screening approaches. This maintains focus on the computational rationale without overstating clinical aspects.

Secondly, Natural product derivatives are ideal for in silico TNBC studies due to their structural diversity, allowing efficient virtual screening and docking. Their known safety profiles and documented biological activity facilitate prioritization and translational potential. Additionally, computational analysis enables rational optimization of functional groups, reducing experimental trial-and-error and accelerating early-stage drug discovery.

Thus, TNBC remains a subtype with limited targeted therapies. Computational screening of natural product derivatives allows rapid evaluation of molecular interactions, electronic properties, and pharmacokinetic potential, providing a cost-effective and rational approach to identify promising candidates. Their inherent structural diversity and generally favorable safety profiles make natural products particularly suitable for early-stage computational drug discovery and subsequent translational studies.

Q4 (1.II). The manuscript seeks to fill a critical knowledge gap on a clinically important topic, but will benefit from a revision across the areas described below.

Response

We sincerely thank the reviewer for recognizing the clinical importance of triple-negative breast cancer (TNBC) and the promise of natural product derivatives as potential therapeutic agents. We have carefully revised the manuscript as per the detailed comments below to improve its scientific clarity, justification, and methodological rigor. We believe the current version more effectively communicates the knowledge gap addressed and the significance of our study.

2. Title

Q5 (2.I). Avoid vague terms like “in silico approaches.” Instead, specify the techniques used. Consider rephrasing for clarity and precision to: “Integrated Quantum Chemical and Molecular Modeling Study of Dihydropyranocoumarins as Potential Inhibitors of Triple-Negative Breast Cancer”

Response

We appreciate the reviewer’s suggestion to enhance clarity and specificity in the title. In accordance with the recommendation, we have revised the title to:

“Integrated Quantum Chemical and Molecular Modeling Study of Dihydropyranocoumarins as Potential Inhibitors of Triple-Negative Breast Cancer.”

This modification replaces the general term “in silico approaches” with a precise description of the methodologies employed, thereby improving the accuracy and scientific rigor of the title.

3. Abstract

Q6 (3.I) Docking against 5HA9 and 7L1X is good, but the biological relevance of these targets to TNBC must be justified in the full text.

Response

We now provide biological roles of these proteins in the Introduction, citing their involvement in TNBC pathways. 5HA9 is a CK2α kinase and 7L1X is a transcription regulator, both implicated in TNBC cell survival and proliferation.

Q7 (3.II) The binding energy of −9.1 kcal/mol is promising, but comparison with known TNBC inhibitors would strengthen the claim.

Response

In the revised manuscript, we have included a comparative analysis of the binding energies of our lead compound with those of reported Triple-Negative Breast Cancer (TNBC) inhibitors from the literature. This comparison, it demonstrates that the observed binding energy (−9.1 kcal/mol) is within or better than the range reported for known TNBC inhibitors, thereby reinforcing the potential efficacy of the proposed compound.

Q8 (3.III) The claim of “high probabilities for antineoplastic activity” needs quantitative backing.

Response

It has been added in the required section as:

PASS (Prediction of Activity Spectra for Substances) was used to evaluate the potential antineoplastic activity of the compounds.

This approach is valuable at the early stages of drug discovery and development. The optimized molecular structures of the target compounds were submitted in MOL file format to the PASS online system, which predicted their potential mechanisms of action and bioactivities. These predictions are based on the probability of activity (Pa) and inactivity (Pi). By applying PASS algorithms and filters, researchers can efficiently screen large libraries of drug candidates, focusing on those with the highest potential, thereby saving time and resources. This strategy can significantly accelerate the drug discovery process and improve the chances of identifying effective treatments for various diseases [52, 53]. The PASS prediction spectrum reveals significant insights into the potential bioactivities of the compounds studied. According to established PASS prediction guidelines, a Pa value> 0.7 is generally considered highly confident, indicating a strong likelihood of biological activity. Values between 0.5 and 0.7 are considered moderate, while those below 0.5 are typically viewed as low-confidence predictions and less likely to be experimentally confirmed.

Q9 (3.IV) The lack of experimental validation is a limitation. Therefore, consider proposing in vitro assays, such as MTT, apoptosis, and migration, as future directions.

Response

We thank the reviewer for this valuable comment. As correctly noted, the present study is entirely in silico and therefore experimental validation was beyond the scope of this work. We fully agree that experimental studies are essential to validate the computational findings. Accordingly, we have now included this point in the revised manuscript as a limitation and suggested that future investigations could involve in vitro assays, such as MTT cytotoxicity, apoptosis, and migration assays, to substantiate the predicted activities.

Q10 (3.V) To improve flow, clarity, and readability, consider simplifying the phrase “...corroborated by MEP maps and charge distribution analyses” and making the phrase “Dynamic compatibility under physiological conditions” more specific, like “stable RMSD and compact protein-ligand complex.”

Response

We thank the reviewer for this insightful suggestion. In response, we have revised the abstract to enhance clarity and precision. Specifically, the phrase “corroborated by MEP maps and charge distribution analyses” has been simplified to “as supported by electrostatic potential and charge distribution analyses,” improving readability while retaining scientific meaning. Additionally, the phrase “dynamic compatibility under physiological conditions” has been replaced with “stable RMSD and compact protein–ligand complex” to more accurately describe the outcomes of the molecular dynamics simulations. These changes have been incorporated in the abstract of the revised manuscript and highlighted by yellow color.

4.Introduction

Q11 (4.I) The introduction is dense and occasionally repetitive. For example, the paragraph beginning with “Therefore, TNBC has fewer treatment options…” repeats earlier points. Consider merging or trimming overlapping content to improve flow.

Response

We thank the reviewer for pointing out the redundancy in the introduction. In the revised manuscript, we have carefully reviewed and streamlined the introduction to remove overlapping statements and improve logical flow. Specifically, the paragraph beginning with “Therefore, TNBC has fewer treatment options…” has been merged with earlier related content to avoid repetition while retaining essential information. This restructuring ensures greater clarity and conciseness for the reader.

Q12 (4.II) The sentence “To the best of our knowledge, we are the first group to report on the anti-cancer properties of coumarins” is factually incorrect since coumarins have been widely studied for anticancer activity. Maybe you should rather say: “To the best of our knowledge, this is the first comprehensive in silico evaluation of dihydropyranocoumarins from Peucedanum japonicum Thunb as potential TNBC therapeutics.”

Response

We appreciate the reviewer’s corrections and helpful suggestions. We agree that coumarins have been extensively studied for anticancer properties, and the original phrasing lacked specificity. Accordingly, we have revised the sentence to: “To the best of our knowledge, this is the first comprehensive in silico investigation of dihydropyranocoumarin derivatives from Peucedanum japonicum Thunb targeting TNBC-associated proteins PARP1 (5HA9) and CK2α (7L1X).”

This revision more accurately reflects the novelty and scope of our study while addressing the reviewer’s concern.

Q13 (4.III) Some references are incomplete or unclear. For example, “World Cancer Day 2021” is not a proper citation. Also, “Bcrf, 2024” should be replaced with a peer-reviewed source or a properly formatted web citation. Ensure that all references are either peer-reviewed or from authoritative databases.

Response

We replaced informal sources with peer-reviewed citations and corrected all reference formatting issues.

Q14 (4.IV) The introduction lacks a clear rationale for the selected TNBC protein targets (e.g., 5HA9 and 7L1X). Consider adding a brief explanation of their biological roles in TNBC (e.g., kinases, transcription factors, etc.).

Response

We appreciate the reviewer’s insightful comment. In response, we have revised the Introduction to include a detailed paragraph explaining the biological relevance of the selected TNBC-associated protein targets PARP1 (PDB ID: 5HA9) and CK2α (PDB ID: 7L1X). We now describe the critical roles of PARP1 in DNA damage repair and synthetic lethality in BRCA-mutated TNBC, and CK2α as a serine/threonine kinase involved in oncogenic pathways (e.g., PI3K/AKT and NF-κB) that promote tumor progression and resistance in TNBC. These additions provide a clear rationale for selecting these targets for our in silico screening.

Q15 (4.V). Several grammatical issues need correction. For example, “This is because TNBC lacks the expression…” can be better phrased as “This is due to the absence of ER, PR, and HER2 expression, rendering hormone and HER2-targeted therapies ineffective.” Also, the phrase “So, the perspe

---

## [Decision Letter · Decision Letter 1]

5 Oct 2025

Evaluation of Dihydropyranocoumarins as Potent Inhibitors Against Triple-Negative Breast Cancer: An Integrated of In silico, Quantum & Molecular Modeling Approaches

PONE-D-25-33604R1

Dear Dr. Kumer,

We’re pleased to inform you that your manuscript has been judged scientifically suitable for publication and will be formally accepted for publication once it meets all outstanding technical requirements.

Kind regards,

Yusuf Oloruntoyin Ayipo, Ph.D

Academic Editor

PLOS ONE

Additional Editor Comments (optional):

The submission is scientifically sound for publication in this title, and all the concerns raised by the respective reviewers regarding the manuscript quality have been satisfactorily addressed. I hereby recommend the manuscript for publication in the current version.

Reviewers' comments:

Reviewer's Responses to Questions

**Comments to the Author**

Reviewer #2: All comments have been addressed

Reviewer #3: All comments have been addressed

2. Is the manuscript technically sound, and do the data support the conclusions?

Reviewer #2: Yes

Reviewer #3: Yes

3. Has the statistical analysis been performed appropriately and rigorously?

Reviewer #2: Yes

Reviewer #3: N/A

4. Have the authors made all data underlying the findings in their manuscript fully available?

Reviewer #2: Yes

Reviewer #3: Yes

5. Is the manuscript presented in an intelligible fashion and written in standard English?

Reviewer #2: Yes

Reviewer #3: Yes

Reviewer #2: The authors have satisfoctoriy addressed the points I raised earlier. The authors have satisfoctoriy addressed the points I raised earlier.

Reviewer #3: Thank you for addressing my comments and that of other reviewers. I recommend that the manuscript is accepted for publication.

**Do you want your identity to be public for this peer review?** For information about this choice, including consent withdrawal, please see our Privacy Policy

Reviewer #2: No

Reviewer #3: No

---

## [Editor Report · Acceptance letter]

PONE-D-25-33604R1

PLOS ONE

Dear Dr. Kumer,

I'm pleased to inform you that your manuscript has been deemed suitable for publication in PLOS ONE. Congratulations! Your manuscript is now being handed over to our production team.

Kind regards,

on behalf of

Dr. Yusuf Oloruntoyin Ayipo

Academic Editor

PLOS ONE